# Two Be or Not Two Be: The Nuclear Autoantigen La/SS-B Is Able to Form Dimers and Oligomers in a Redox Dependent Manner

**DOI:** 10.3390/ijms22073377

**Published:** 2021-03-25

**Authors:** Nicole Berndt, Claudia C. Bippes, Irene Michalk, Dominik Bachmann, Jennifer Bachmann, Edinson Puentes-Cala, Tabea Bartsch, Liliana R. Loureiro, Alexandra Kegler, Ralf Bergmann, Joanne K. Gross, Tim Gross, Biji T. Kurien, R. Hal Scofield, A. Darise Farris, Judith A. James, Marc Schmitz, Karim Fahmy, Anja Feldmann, Claudia Arndt, Michael P. Bachmann

**Affiliations:** 1Department of Radioimmunology, Institute of Radiopharmaceutical Cancer Research, Helmholtz-Zentrum Dresden-Rossendorf (HZDR), 01328 Dresden, Germany; n.berndt@hzdr.de (N.B.); epuentes@corrosion.uis.edu.co (E.P.-C.); t.bartsch@hzdr.de (T.B.); l.loureiro@hzdr.de (L.R.L.); a.kegler@hzdr.de (A.K.); r.bergmann@hzdr.de (R.B.); a.feldmann@hzdr.de (A.F.); c.arndt@hzdr.de (C.A.); 2Institute of Immunology, Medical Faculty Carl Gustav Carus Dresden, Technical University Dresden, 01307 Dresden, Germany; claudia_bippes@gmx.de (C.C.B.); Irene.Michalk@uniklinikum-dresden.de (I.M.); marc.schmitz@tu-dresden.de (M.S.); 3University Cancer Center (UCC), Tumor Immunology, University Hospital Carl Gustav Carus Dresden, Technical University Dresden, 01307 Dresden, Germany; d.bachmann@posteo.de (D.B.); bynnej@gmail.com (J.B.); 4Corporación para la Investigación de la Corrosión (CIC), Piedecuesta 681011, Colombia; 5Department of Biophysics and Radiobiology, Semmelweis University, 1094 Budapest, Hungary; 6The Arthritis and Clinical Immunology Program, Oklahoma Medical Research Foundation and University of Oklahoma Health Sciences Center, Oklahoma City, OK 73104, USA; Jody-Gross@omrf.org (J.K.G.); Tim-Gross@omrf.org (T.G.); Biji-Kurien@omrf.org (B.T.K.); hal-scofield@omrf.org (R.H.S.); Darise-Farris@omrf.org (A.D.F.); Judith-James@omrf.org (J.A.J.); 7National Center for Tumor Diseases (NCT), 01307 Dresden, Germany; 8Institute of Resource Ecology, Helmholtz-Zentrum Dresden-Rossendorf (HZDR), 01328 Dresden, Germany; k.fahmy@hzdr.de

**Keywords:** anti-La/SS-B antibodies, autoimmunity, La/SS-B autoantigen, systemic lupus erythematosus, primary Sjögren’s syndrome

## Abstract

According to the literature, the autoantigen La is involved in Cap-independent translation. It was proposed that one prerequisite for this function is the formation of a protein dimer. However, structural analyses argue against La protein dimers. Noteworthy to mention, these structural analyses were performed under reducing conditions. Here we describe that La protein can undergo redox-dependent structural changes. The oxidized form of La protein can form dimers, oligomers and even polymers stabilized by disulfide bridges. The primary sequence of La protein contains three cysteine residues. Only after mutation of all three cysteine residues to alanine La protein becomes insensitive to oxidation, indicating that all three cysteines are involved in redox-dependent structural changes. Biophysical analyses of the secondary structure of La protein support the redox-dependent conformational changes. Moreover, we identified monoclonal anti-La antibodies (anti-La mAbs) that react with either the reduced or oxidized form of La protein. Differential reactivities to the reduced and oxidized form of La protein were also found in anti-La sera of autoimmune patients.

## 1. Introduction

Almost five decades ago, autoantibodies to the autoantigens La and Ro, also known as Sjögren’s syndrome associated antigen B (SS-B) and SS-A, were first detected in sera of patients suffering from primary Sjögren’s syndrome (pSS) or systemic lupus erythematosus (SLE) [1,2]. Since then, a series of in part controversial data was published about the structure, function and localization of both autoantigens. As part of the diagnosis, sera of patients are commonly analyzed for the presence of anti-nuclear antibodies (ANAs) using immunofluorescence (IF) microscopy. According to these countless microscopical analyses, there is certainly no doubt about a nuclear localization of La protein [3]. Nuclear localization is also in good agreement with the described nuclear function(s) of La protein in the context of primary RNA polymerase III transcripts [4,5,6,7,8,9]. Besides these nuclear functions, however, it was published that La protein may play an additional role in the cytoplasmic compartment: It was shown that La protein is involved in Cap-independent translation of viral and cellular mRNAs harboring internal ribosomal entry sites (IRES elements) [8,9,10,11,12,13]. Using Far-Western blotting, Craig et al. identified a dimerization domain in La protein and showed experimental evidence that dimerization is required for La protein’s function in IRES-dependent translation [14]. The dimerization element was identified in the C-terminal domain of La protein between amino acids (aa) 293–348. Later, a nuclear retention element (NRE) and a nucleolar localization signal (NoLS) were identified inside of the same region [15,16] (Figure 1A, and also below).

The mobility of recombinantly expressed human La protein as estimated by SDS–PAGE is in good agreement with its theoretical molecular weight of 50 kDa. Using size-exclusion gel filtration instead, a native molecular weight of around 100 kDa can be estimated [14], which would be in line with the existence of a La protein dimer. However, the deviation from the theoretical molecular weight was interpreted as the result of aberrant mobility caused by a non-globular but rod-like structure of native La protein [14]. Unfortunately, even today, the structure of full-length La protein has not been completely clarified. Only the structure of three fragments of La protein, the N-terminal La motif (aa1-103), the RNA-binding domain 1 (RRM1) (aa105-202) and the RRM2 (aa225-334) domain were obtained at high-resolution [15,17,18,19,20] (see also Figure 1B). According to these structural studies, La protein should not form dimers. The monomeric structure of La protein was supported by an ultracentrifugation analysis [15]. However, it is important to mention that crystallization and ultracentrifugation analysis was performed under reducing conditions [15].

In completely independent studies, Huth et al. searched for a protein-based substrate for a toxicological evaluation of thiol-reactive compounds [21]. Unexpectedly, an internal portion of the La protein (aa100-324) was identified as the most suitable substrate for such an assay, which was termed as ALARM NMR [21,22,23]. The abbreviation ALARM NMR stands for a La assay to detect reactive molecules by nuclear magnetic resonance. For this assay, the authors use a fragment of La protein, which starts downstream of the La motif (aa1-105) and, thus, lacks the cysteine residue Cys18 in the α1-helix of the La motif. The redox reaction of the ALARM NMR assay is based on the two cysteine residues (Cys232 and Cys245) present in the C-terminal RRM2 domain of La protein (Figure 1A). Obviously, the two cysteine residues (Cys232 and Cys245) are not part of the dimerization domain (aa293-348) as they locate upstream of it (Figure 1A, gray bar).

Interestingly, all cysteine residues in La protein are highly conserved during evolution, including the cysteine residue Cys18 in the La motif of the La-related protein 4 (LARP4), which is a poly(A)-binding protein [17].

According to the ALARM NMR studies, the La fragment used in this assay is sensitive to oxidation and undergoes redox-dependent conformational changes [21,22,23]. However, it remains unclear whether or not this is also true for native full-length La protein. Furthermore, no data are available whether or not all three cysteine residues or just the two cysteine residues Cys232 and Cys245, contribute to the redox sensitivity. Moreover, it remains open if oxidation of one or the other cysteine(s) has different structural and functional consequences as, for example, in the case of the high mobility group protein 1 (HMGB1) [24]. It is commonly known that cysteine residues can undergo a series of different oxidations, which can modulate the function of proteins [25]. Both intra- and intermolecular disulfide bridges can be formed and function as redox-sensing molecular switches responding to oxidative stress. The function of such redox-sensing proteins ranges from enzymes, transcriptional factors or modulators, receptor proteins to sensor proteins [26,27,28,29,30,31]. In principle, there are two kinds of regulation mechanisms possible: (i) a direct functional modulation via locking and unlocking of a critical cysteine residue, and (ii) an indirect modulation via a conformational change, which affects the function of a distantly located region in an allostery-like manner [25].

Interestingly, the nuclear HMGB1 shares many similarities with La protein [24]. Both proteins have been described as alarmins having additional functions in the immune system besides their nuclear function [32]. Like La protein, HMGB1 contains three cysteine residues (Cys23, Cys45, and Cys106). These cysteine residues are sensitive to oxidation. During purification, HMGB1 forms both homodimers and oligomeric forms of the protein. Cys106-mediated formation of HMGB1 dimers was also observed under conditions of excessive reactive oxygen species (ROS) generation. Oxidation of HMGB1 results in intramolecular disulfide bonds depending on the ROS concentration [33]. As also described for La protein [34,35,36,37,38], oxidative stress results in a translocation of HMGB1 from the nucleus to the cytoplasm [24]. Furthermore, like La protein [39,40], HMGB1 can be released from cells and bind to the extracellular matrix [24]. In dependence on their redox status, extracellular forms of HMGB1 seem to have different functions: Fully reduced HMGB1 can induce autophagy through binding to RAGE. In the case Cys23 and Cys45 are oxidized, HMGB1 is able to signal through TLR4 and can lead to proinflammatory cytokine release [24]. As HMGB1 is elevated in sera of SLE patients and levels of serum HMGB1 correlate with disease activity [24], HMGB1 is assumed to play a role in these autoimmune diseases. Until now, it is not known if La protein shows similar redox-dependent features and functions.

Here we present evidence that full-length La protein is sensitive to oxidation. All three cysteine residues are involved in the oxido-reduction of La protein. Using biophysical approaches, we show that La protein undergoes redox-dependent conformational changes. Moreover, we provide evidence for the existence of anti-La Abs present in autoimmune patient sera, which preferentially react with either the oxidized or reduced form of La protein. Furthermore, we describe monoclonal antibodies (mAbs) that can differentiate between the oxidized and reduced form of La protein. In addition, we show that oxidation of La protein leads to the formation of dimers, oligomers, and even insoluble polymers, which are stabilized by disulfide bridges. While the dimerization domain in the RRM2 is cryptic under reducing conditions, it becomes accessible after oxidation. Besides the dimerization domain in the RRM2, we identified a further protein interaction site in the N-terminal La motif of La protein.

## 2. Results

### 2.1. The Accessibility of the Epitope Recognized by the Anti-La mAb 7B6 Is Dependent on Oxidation

The hybridoma secreting the anti-La mAb 4B6 was first described in 1995 [46]. To maintain its productivity, the hybridoma had to be recloned several times. Thereby subclones were obtained. One of the resulting subclones is the anti-La mAb 7B6. The original cloning and reclonings were screened by ELISA using recombinantly expressed human La protein. As will be shown below, recombinantly expressed La protein is a mixture of the oxidized and reduced forms of La protein, which explains how the anti-La mAb 4B6/7B6 was identified in the first place.

Since its first description, the anti-La mAb 7B6 was used in different labs. While it reliably worked in ELISA and Western blotting experiments, controversial results were reported for IF studies. The reason for these discrepancies remained unclear for many years. The first evidence came by chance: We used the anti-La mAb 7B6 to teach medical students how to perform IF microscopy. For that purpose, several hundred coverslips were prepared, fixed with methanol, and stained in parallel. Usually, we store the methanol for fixation in plastic bottles. In this case, the amount of methanol was not sufficient for the fixation of all coverslips. Therefore, a minor portion of the coverslips was fixed using methanol of analytical grade that was taken from a freshly opened dark brown glass bottle. As expected, human cells fixed with methanol stored in plastic bottles gave nuclear staining in IF microscopy (Figure 2A,B). However, the same batch of mAb failed to stain fixed cells (Figure 2C, lower panel) if the methanol for fixation was taken from a freshly opened dark brown glass bottle (Figure 2D). Bearing in mind the reported sensitivity of the La protein fragment to oxidation [21,22,23], we speculated that the methanol stored in the plastic bottle could contain, e.g., peroxides or oxygen radicals, which may be absent in the methanol of analytical grade stored in the dark and in the brown glass bottle. Indeed, when the same coverslips that failed in the first staining round were shortly rinsed with PBS, to which hydrogen peroxide was added (Figure 2E), the staining could be restored (Figure 2F, lower panel). Already a single drop of hydrogen peroxide solution added to 15 mL PBS (equivalent to about 30 μM H_2_O_2_) in the staining chamber was sufficient.

From this result, we concluded that the accessibility of the epitope recognized by the anti-La mAb 7B6 is cryptic under reducing conditions but becomes accessible under oxidative conditions.

### 2.2. The 7B6 Epitope Is Part of the Dimerization Domain, the Nuclear Retention Element, and the Nucleolar Localization Signal

In order to learn how oxidation can influence the accessibility of the 7B6 epitope, epitope mapping studies were performed. As described previously, for epitope mapping, we used a series of deletion mutants of human La protein [45]. The La fragments were truncated from either the N- or the C-terminus or from both sites. All deletion mutants were expressed in *E. coli* as 6xHis tagged proteins and purified by nickel affinity chromatography. The purified La fragments were separated by SDS–PAGE, and immunoreactivity was determined by immunoblotting. Selected data for the most relevant La fragments are summarized in Figure 3A–D. According to these data, the smallest N- and C-terminally truncated La fragment, which still reacts with the anti-La mAb 7B6 consists of the aa sequence EKEALKKIIEDQQESLNK (aa311-328) of human La protein.

The identified epitope represents the α3 helix in the RRM2 domain in the C-terminal portion of the La protein (Figure 3C); see also [15,17,18,19,20]. It is therefore, part of previously described important functional elements, including the dimerization domain (Figure 1A and Figure 3A, Dim, aa293-348), the nuclear retention element (Figure 3A, NRE, aa316-332) and the nucleolar localization signal (Figure 3A, NoLS, aa323-354) [15,17,18,19,20].

### 2.3. Further Characterization of the Epitope Sequence Recognized by the Anti-La mAb 7B6

In previous studies, the two cysteine residues (Cys232 and Cys245, see also Figure 1A) in the RRM2 domain (Figure 1A,B) of La protein were found to be highly sensitive to oxidation [21,22,23] (see also the introduction). Obviously, they are not part of the identified 7B6 epitope sequence (aa311-328). The isolated peptide sequence is most likely not highly sensitive to oxidation (EKEALKKIIEDQQESLNK). It remains, however, open whether or not the carboxyl group of the glutamate residue Glu311 in the epitope sequence recognized by the anti-La mAb 7B6 (EKEALKKIIEDQQESLNK) highlighted in green, which is in close vicinity to the SH group of cysteine residue Cys232 in the RRM2 domain can form an intramolecular thioester bond. Such an internal thioester bond would most likely be highly reactive.

The human 7B6 epitope sequence EKEALKKIIEDQQESLNK (Figure 4A, hLa) differs from the murine epitope sequence EKEALKKITDDQQESLNK (Figure 4A, mLa) with respect to the two aa (IE > TD) highlighted in red. As the murine La protein expressed in *E. coli* reacts with the anti-La mAb 7B6, while the murine La protein expressed in eukaryotic cells fails to react, we assume that one or both aa residues (TD) undergo posttranslational modification(s) in mouse cells. In order to confirm this assumption, we cloned N-terminally tagged 7B6 epitope/enhanced green fluorescent protein (EGFP) fusion proteins with (Figure 4A, E-EGFP-NLS) and without a C-terminal nuclear localization signal (Figure 4A, E-EGFP). The resulting constructs were transfected into either human HEK293 cells and analyzed by SDS–PAGE and immunoblotting (Figure 4B) or human HeLa cells and analyzed by IF microscopy (Figure 4C,D). As a negative control, we used EGFP and the respective EGFP-NLS vector lacking the N-terminal epitope sequence. As summarized in Figure 4B–D the data obtained by both techniques for the different human cell lines are comparable, indicating that the results do neither depend on the cell line nor the different analytical techniques.

In detail, after transfection of HEK293 cells, total extracts were prepared and analyzed by SDS–PAGE and immunoblotting (Figure 4B). All extracts were tested against anti-EGFP (Figure 4B, upper blot, anti-GFP) and against the anti-La mAb 7B6 (Figure 4B, lower blot, anti-La 7B6). EGFP and all EGFP fusion constructs could be detected in the respective extract (Figure 4B, upper blot, anti-GFP). Thus, the HEK293 cells were efficiently transfected, and the respective transgene comparably well expressed. As expected, all the HEK293 cell extracts contain endogenous human La protein, which can be detected by the anti-La mAb 7B6 (Figure 4B, lower blot, anti-La 7B6). The reactivity of anti-La 7B6 mAb to endogenous La protein was used as the loading control. Obviously, the mobility of the endogenous La protein differs from the mobility of the respective EGFP fusion proteins. Thus, the EGFP-fusion proteins can easily be separated from the endogenous La protein (Figure 4B, lower blot, anti-La 7B6). As also expected, the extracts from cells transfected with plasmids encoding EGFP or EGFP-NLS reacted with the anti-EGFP Ab (Figure 4B, upper blot, anti-GFP, lanes GFP and GFP-NLS), while the anti-La mAb 7B6 does not recognize the untagged either EGFP or EGFP-NLS construct (Figure 4B, lower blot, anti-La 7B6, lanes GFP and GFP-NLS). As schematically summarized in Figure 4A, the fusion of the human or murine 7B6 epitope sequence or mutant variants to the N-terminus of EGFP or EGFP-NLS increases their molecular weight (Figure 4B). After fusion of the human 7B6 epitope, the resulting epitope-tagged EGFP and EGFP-NLS fusion proteins reacted with both the anti-EGFP Ab (Figure 4B, upper blot, anti-GFP, lanes IE) and the anti-La mAb 7B6 (Figure 4B, lower blot, anti-La 7B6, lanes IE). Thus, the 7B6 epitope, when used as a protein tag, is accessible for the anti-La mAb 7B6 irrespective of its localization in the cytoplasm (Figure 4B, lanes IE, -NLS) or the nucleus (Figure 4B, lanes IE, +NLS). These data are confirmed by the IF studies (Figure 4C,D). The untagged EGFP-NLS fusion protein is predominantly found in the nucleus (Figure 4C, GFP), while EGFP without the NLS is predominantly found in the cytoplasm (Figure 4D, GFP). The partial nuclear localization of EGFP lacking the NLS may be caused by diffusion or a week intrinsic NLS in EGFP, as reported earlier [47]. Only the transfected cells expressing the human 7B6 epitope EGFP (Figure 4D, IE) or EGFP-NLS (Figure 4C, IE) fusion proteins were stained with the anti-La mAb 7B6, while the endogenous human La protein was not accessible under the selected fixation condition. This becomes evident when comparing the DAPI staining pattern with the staining of the anti-La mAb 7B6 (Figure 4C,D). For reliability, at least ten areas showing at least 50 cells were analyzed per coverslip. The respective combined EGFP/Ab-staining pattern did not differ. According to these data, the isolated 7B6 sequence itself seems not to be dependent on oxidation. Furthermore, the 7B6 epitope is cryptic in endogenous human nuclear La protein but becomes accessible under oxidative conditions.

### 2.4. The 7B6 Epitope Undergoes Posttranslational Modifications

The human 7B6 sequence contains isoleucine at aa position 319 and glutamic acid at aa position 320. In the murine sequence, these two aa are replaced by threonine and aspartic acid, respectively (see also Figure 1A and Figure 4A, hLa, mLa). Bearing in mind that the anti-La mAb 7B6 reacts with mouse La protein if recombinantly expressed in *E. coli* but fails to react with mouse La protein if it is present in eukaryotic cells [45] (and also Figure 4C,D), we concluded that one or both aa should undergo a posttranslational modification. In order to confirm this conclusion, we prepared further La epitope EGFP fusion mutants. We replaced either the isoleucine with threonine (Figure 4B–D, TE) or the glutamic acid with aspartic acid (Figure 4B–D, ID) or both, the isoleucine and the glutamic acid with threonine and aspartic acid (Figure 4B–D, TD). All these La epitope variants were fused to the N-terminus of EGFP and also to EGFP-NLS and used for transfection of the human cell lines HEK293 and HeLa. Extracts from transfected HEK293 cells were prepared and analyzed by SDS–PAGE and immunoblotting (Figure 4B). HeLa cells were fixed and analyzed by IF microscopy (Figure 4C,D). Replacing the aa IE with TD completely abolishes the reactivity in SDS–PAGE and immunoblotting (Figure 4B, lanes TD) as well as in IF microscopy (Figure 4C,D, TD). Replacement of the glutamate with aspartate (Figure 4B–D, ID) had no effect on the reactivity in either SDS–PAGE and immunoblotting (Figure 4B, lanes ID) or IF microscopy (Figure 4C,D, ID). In contrast, the TE mutant did not react, especially when localized in the nucleus either by SDS–PAGE/immunoblotting (Figure 4B, lane TE, +NLS) or IF microscopy (Figure 4C, TE). There is a faint band detectable with the anti-La mAb 7B6 when this mutant is expressed in the cytoplasm (Figure 4B, lane TE, -NLS), which is hardly detectable by IF microscopy (Figure 4D, TE).

From these results, we conclude, the failure of the anti-La mAb 7B6 to react with the murine La protein is due to a posttranslational modification of the threonine residue, which replaces the aa isoleucine in the corresponding human epitope sequence. This posttranslational modification exists in both compartments, the nucleus and the cytoplasm. In the nucleus most if not all La molecules should be modified, while the modification is less efficient in the cytoplasm or partially removed.

It is tempting to assume that the threonine residue in the mouse sequence becomes phosphorylated especially bearing in mind that the serine at position 325 in the human 7B6 epitope sequence has already been reported to undergo posttranslational phosphorylation [48]. In this study, phosphorylation of La protein was analyzed in detail and shown that it does not influence the localization of La protein [48]. In order to analyze the influence of such a posttranslational modification on the reactivity of the anti-La mAb 7B6, we constructed four additional epitope mutants. We replaced serine 325 in the human sequence with either aspartate or alanine, thus simulating either permanently phosphorylated or non-phosphorylated serine 325. As shown by both techniques SDS–PAGE and immunoblotting (Figure 4B, lanes S > D) and IF microscopy (Figure 4C,D, S > D), simulation of permanent phosphorylation of serine 325 completely abolishes the reactivity of the anti-La mAb 7B6. In contrast, the anti-La mAb 7B6 is able to react with the serine to alanine mutant both in SDS–PAGE and immunoblotting (Figure 4B, lanes S > A) and IF microscopy (Figure 4C,D, S > A).

From these results, we conclude that the human epitope sequence can undergo a posttranslational modification, most likely phosphorylation of serine 325, which abolishes the reactivity of the epitope with anti-La mAb 7B6. As the anti-La mAb 7B6 reacts with the epitope EGFP fusion proteins after SDS–PAGE and immunoblotting as well as IF microscopy, serine 325 cannot completely be posttranslationally modified in human nuclear La protein.

In summary, the human epitope sequence, which is recognized by the anti-La mAb 7B6, may not be sensitive to oxidation. However, oxidation should somehow affect the accessibility of the epitope. Besides oxidation, a posttranslational modification at serine 325 can influence the reactivity of the anti-La mAb. Although not finally proven, this posttranslational modification is most likely phosphorylation as the replacement of the serine by an aspartate, which simulates permanent phosphorylation blocks the reactivity of the anti-La mAb 7B6. Consequently, when interpreting immunoblotting or IF data, one has always to keep in mind that the anti-La mAb 7B6 detects only a subpopulation of human La protein, namely the unmodified human La protein, and that the epitope is cryptic under reducing conditions. The replacement of the isoleucine and glutamate by threonine and aspartate in the murine La sequence allows a posttranslational modification again, most likely phosphorylation of the murine La protein, which blocks the reactivity of the anti-La mAb 7B6.

### 2.5. The 7B6 Epitope Representing the α3−Helix of the RRM2 Domain of La Protein Alone Is Not Sufficient for Dimerization

As mentioned in the introduction section, La protein contains a controversially discussed dimerization domain, which is part of the RRM2 domain of La protein and consists of the aa293-348 (Figure 1A and Figure 5A). Consequently, the 7B6 epitope is part of this potential dimerization domain. The dimerization domain was originally detected using a protein–protein (Far-Western) blotting approach [14]. For this purpose, La protein deletion mutants were blotted onto a membrane and incubated with a radioactively labeled La protein probe. Protein–protein interactions were detected by autoradiography.

In order to learn whether or not the 7B6 epitope representing the α3-helix of the RRM2 domain of La protein is sufficient for dimerization, we performed a comparable experiment. However, we modified the Far-Western blotting assay as follows: After blocking, the membrane was incubated with unlabeled full-length La protein. The interaction of the blotted La protein fragments with native La protein was identified using the anti-La mAb SW5. SW5 recognizes a conformational epitope consisting of N- and C-terminal portions of the RRM1 domain (aa112–138 and aa171–183) [49,50]. Therefore, all the deletion mutants, which were included in this study started downstream of the SW5 epitope. They were expressed as 6xHis-tagged proteins in *E. coli*. After SDS–PAGE and immunoblotting, the selected deletion mutants could, therefore, be detected by an anti-His Ab (Figure 5B, anti-His). Besides the respective La fragment of interest, additional fragments according to lower molecular weight were detected in some of the La protein preparations. Their origin is not completely clear. Most likely, they represent proteolytic degradation products. An alternative explanation could be: La protein contains internal methionine residues where internal initiation of the translation may occur. In agreement with the described conformational epitope recognized by the anti-La mAb SW5, the anti-La mAb SW5 reacted with full-length La protein (Figure 5B, anti-La SW5, lanes 1–408), but not with any of the N-terminally truncated La proteins (Figure 5B, anti-La SW5) or these fragments. When the blot was incubated with full-length La protein and then tested for anti-La SW5 reactivity all the selected La deletion mutants, but the human 7B6 epitope became reactive with the anti-La mAb SW5 supporting the interaction of full-length La protein with these La deletion mutants (Figure 5B, La + anti-La SW5). Thus, our data confirm the previously reported presence and localization of a dimerization domain in La protein. We found that the deletion mutant consisting of aa303-344 still allows such an interaction. Consequently, the dimerization domain can be further limited to this region. As also evident from our data, the 7B6 epitope itself is not sufficient for dimerization but part of the protein–protein interaction site (Figure 5B, La + anti-La SW5, lanes 311–328).

The following experiment was originally planned as a control experiment but turned out with an unexpected result. Using the same Far-Western blotting strategy, we tested a C-terminally truncated La fragment (Figure 5A, LaN) consisting of aa1-192 and thus representing the La motif plus the RRM1 domain of La protein (see also Figure 1A) for binding to LaC, which starts with the RRM2 domain and includes aa194-408. As schematically shown in Figure 5A, LaC contains the dimerization domain. In agreement, it interacts with full-length La protein as expected (Figure 5B, La +anti-La SW5, lanes 194–408). As the LaN fragment lacks the downstream located dimerization domain, we expected that it would not bind to LaC. However, it interacts with LaC (Figure 5C, lane LaN) like full-length La protein (Figure 5C, lane La wt). To further support this result, the interaction was confirmed using instead of the anti-La mAb SW5 the anti-La mAb 5B9 (Figure 5C, anti-La 5B9). The anti-La mAb 5B9 recognizes a short peptide epitope sequence of the 10 aa KPLPEVTDEY (aa95–104) of human La protein, which is the linker between the La motif and the RRM1 domain [51]. The experiment using the anti-La mAb 5B9 confirmed the results with the anti-La mAb SW5. According to these data, LaN should contain an additional protein interaction domain, which was not detected in the previous study [14]. If so, the LaN domain could bind internally to the LaC domain, thereby hiding the dimerization domain, the NRE and/or the NoLS, and perhaps the 7B6 epitope in a redox-dependent manner. Theoretically, besides this monomeric intramolecular interaction, in an opened form, theoretically, LaN may also be able to interact with LaC of another La molecule, thus forming different forms of dimers, including head–head/tail-tail dimers (Figure 5DI), but also head to tail dimers (Figure 5DII). Furthermore, a variety of oligomers and even polymers could be formed based on LaN-LaN, LaC-LaC or LaC-LaN interactions between different La molecules (Figure 5DIII).

In order to rule out that denaturing of La-related fragments may allow interactions with hydrophobic regions [15], we decided to confirm the Far-Western blotting data by surface plasmon resonance (SPR) studies (Figure 6). For this purpose, full-length human La protein (La1-408) was covalently linked to a sensor plate. Bovine serum albumin served as control (Figure 6, BSA). Sensograms for selected La protein deletion mutants are shown in Figure 6: All La fragments containing the Dim domain are highlighted in green. All La fragments lacking the Dim domain are highlighted in red. As shown in Figure 6, native full-length La protein (La 1–408) binds to full-length La protein confirming the protein–protein interaction. Moreover, the N-terminally truncated fragments La 158–408, La 239–408, La 278–408, La 303–408 (all containing the Dim domain) nicely bind to full-length La protein. In contrast, the La fragment La 348–408 starting C-terminally of the Dim domain fails to bind. These protein–protein interactions should be specific as the fragments do not bind to BSA, as shown in Figure 6 for the La deletion mutants La 278–408 and La 1–371, both containing the Dim domain. As also found by the Far-Western blotting experiment, both C-terminally truncated La deletion mutants containing (La 1–371) or lacking the Dim domain (La 1–284, and La 1–245) are capable of protein–protein interactions.

### 2.6. La Protein Forms Dimers and Oligomers That Can Be Stabilized by Disulfide Bridges

According to the published ALARM NMR assay [21,22,23], the two C-terminal cysteine residues (Cys232 and Cys245) in the RRM2 domain of La protein are sensitive to oxidation (see also the introduction). The here presented data support the idea that the cysteine residue Cys18 in the La motif could also form a disulfide bridge with one of the C-terminal cysteine residues (Cys232 and Cys245).

During storage of La protein, we frequently observed that recombinantly expressed wild-type La protein (Figure 7A1) tends to precipitate, including at 4 °C, which can be prevented by the addition of ß-mercaptoethanol or dithiothreitol (DTT) to the purified La protein batches (data not shown). Mutating the three cysteine residues to alanine residues affects the formation of these aggregates dramatically (Figure 7A2). Besides the triple cysteine mutant, we, therefore, prepared further La mutants. We either replaced the two cysteine residues (Cys232 and Cys245) in the RRM2 domain resulting in the double cysteine mutant as schematically shown in Figure 7B3 or the cysteine residue (Cys18) in the La motif resulting in the mono cysteine mutant as schematically shown in Figure 7B4.

After recombinant expression and purification, the four isolated protein samples were analyzed by SDS–PAGE and immunoblotting (Figure 7C, wild-type La protein (1) lanes 1, triple cysteine mutant (2) lanes 2, mono cysteine mutant (3) lanes 3, double cysteine mutant (4) lanes 4). For detection of the La-related protein bands, we used the anti-La mAb SW5 (Figure 7C, upper blots) or the anti-La mAb 7B6 (Figure 7C, lower blots). The protein samples for SDS–PAGE were either prepared in the absence of ß-mercaptoethanol (Figure 7C, untreated) or in the absence of ß-mercaptoethanol, but treated with CuSO_4_ (2 mM) prior to the sample preparation (Figure 7C, oxidized), or in the presence of ß-mercaptoethanol (Figure 7C, reduced). In all protein samples prepared under reducing conditions, only the monomeric La protein can be detected with both anti-La mAbs (Figure 7C, reduced, lanes 1 to 4). In contrast, all untreated or oxidized protein samples containing either the three cysteine residues (Figure 7C, untreated, oxidized, lanes 1) or the two cysteine residues (Cys232 and Cys245) (Figure 7C, untreated, oxidized, lanes 3), or the single cysteine residue (Cys18) (Figure 7C, untreated, oxidized, lanes 4) additional anti-La reactive protein bands with lower mobilities according to molecular weights of dimers or even higher oligomers can be detected. When comparing the immunoblots of untreated (Figure 7C, untreated, lanes 1, 3, 4) with oxidized proteins (Figure 7C, oxidized, lanes 1, 3, 4), it becomes evident that the intensity of these higher molecular weight protein bands increases in the case the samples are pretreated with CuSO_4_ prior to SDS–PAGE. Most importantly, these higher molecular weight anti-La reactive protein bands are absent in the triple cysteine mutant, including after oxidation (Figure 7C, untreated, oxidized, lanes 2). Consequently, these anti-La reactive higher molecular weight protein bands contain disulfide bridges that can be reduced, leading to monomeric La protein. In the triple cysteine mutant, such disulfide bridges cannot be formed, which is the reason why the La-related high molecular weight bands are absent in these samples (Figure 7C, untreated, oxidized, reduced, lanes 2). In addition, important to mention both wild-type La and the mono cysteine mutant, which consequently contains the remaining two cysteine residues (Cys232 and Cys245) in the RRM2 domain, can form higher aggregates than dimers (Figure 7C, untreated, oxidized, lanes 1, 3). In contrast, the double cysteine mutant in which only the single N-terminal cysteine residue Cys18 remains can still form dimers, but no higher oligomers (Figure 7C, untreated, oxidized, lanes 4). According to these data, not only the cysteine residues Cys232 and Cys245 in the RRM2 domain, but also the N-terminal cysteine residue Cys18 is sensitive to oxidation and can form disulfide bridges supporting that oxidation of La protein may lead to all the hypothesized dimers and oligomers, as illustrated in Figure 5D. In order to further validate the immunoblotting data, the blots were densitometrically evaluated. For this purpose, the individual lanes were scanned. The scans are shown in Appendix A. The densitometric data clearly support the eyeball estimated interpretations that the nickel affinity isolated La protein preparations already contain dimers and oligomers besides the monomeric La protein. Moreover, these dimers disappear when the samples are reduced prior to SDS–PAGE and immunoblotting. Furthermore, exposure of wild-type La protein to oxidative conditions increases the number of dimers and oligomers. Mutation of all three cysteine residues avoids the formation of these dimers and oligomers.

As mentioned above and schematically shown in Figure 8A,B, the cysteine residue (Cys232) locates close to the glutamate (Glu311) of the 7B6 epitope. Moreover, the cysteine residue (Cys232) locates at the N-terminal site of a ß-sheet (Figure 8B), while the cysteine residue (Cys245) locates downstream of the ß-sheet and a less structured sequence of aa (Figure 8B). Thus, the two cysteine residues locate almost diagonal in the RRM2 domain (Figure 8A, arrows). In order to form a disulfide bridge, both cysteines must be brought in close vicinity, which may alter the structure of the RRM2 domain, as also observed in the ALARM NMR studies for the analyzed La fragment. This should affect the accessibility of the α3-helix. Alternatively, the glutamate Glu311 is located very close to the cysteine residue Cys232 in the RRM2 domain (Figure 8A), leading to the speculation that the side-chains of these two aa could perhaps form a highly reactive internal thioester in a redox-dependent manner.

### 2.7. Antibodies Recognizing the Reduced Form of La Protein

So far, our data show that the epitope recognized by the anti-La mAb 7B6 is directed to an epitope that becomes accessible under oxidative conditions. It would be of special interest to have anti-La mAbs, which specifically recognize the reduced nuclear form of La protein. Such an antibody pair could become a useful tool to follow redox-dependent conformational changes and the resulting effects on the function and localization of La protein. Recently, six novel anti-La mAbs were described by us [45]. Two of this anti-La mAbs (22A and 24G7) were shown to be directed against the RRM1 domain. The other four anti-La mAbs (2F9, 312B, 32A and 27E) were directed to the La motif, which contains the cysteine residue Cys18. According to epitope mapping, all of this anti-La mAbs recognize conformational epitopes [45]. In order to differentiate between reactivity to the reduced or oxidized form, we established an ELISA comparing the reactivity of wild-type La versus the triple cysteine mutant both prior or after treatment with 2 mM CuSO_4_. According to the immunoblotting data, we expected that only wild-type La protein is sensitive to oxidation under these conditions, while the triple cysteine mutant should not be influenced by oxidation. Indeed, as shown in Figure 9A, oxidation of wild-type La protein completely abolishes the reactivity of all the four anti-La mAbs 2F9, 312B, 32A and 27E recognizing the La motif (Figure 9A, black bars), while oxidation of the triple cysteine mutant has no effect on their reactivity (Figure 9A, gray bars). In contrast, there was no effect on the reactivity of mAbs that are directed to the RRM1 domain (Figure 9A, SW5, 22A, 24G7), the randomly coiled linker region between the La motif and the RRM1 (Figure 9A, 5B9) or the dimerization site (Figure 9A, 7B6). Interestingly, when a mixture of Abs is analyzed, the presence of an Ab specific for the reduced form can easily be overlooked (Figure 9A, 5B9 + SW5 + 2F9). The data obtained by ELISA were confirmed by SDS–PAGE and immunoblotting (Figure 9B). Samples of wild-type La protein were analyzed by SDS–PAGE/immunoblotting and tested against the anti-La mAb 2F9, one of the mAbs, which we found to preferentially react with the reduced form of La protein. Prior to SDS–PAGE and immunoblotting, the samples were either reduced (Figure 9B, sample reduced) or not (Figure 9B, sample not reduced). In agreement with the ELISA data, the anti-La mAb 2F9 recognizes the reduced La protein. In contrast, it failed to react with the La sample, which was not reduced prior to electrophoresis (Figure 9B, sample not reduced, membrane not reduced). Thereafter, the latter blot was cut into two halves. The right portion was rinsed in PBS containing ß-mercaptoethanol, washed and tested again for reactivity to the anti-La mAb 2F9. Now the anti-La mAb 2F9 was able to detect La protein (Figure 9B, sample not reduced, membrane reduced, -La). As the sample was not reduced prior to electrophoresis, it contained an oxidized dimer of La protein, which after reduction on the blot could also be detected by the anti-La mAb 2F9 (Figure 9B, sample not reduced, membrane reduced, -La dimer).

In order to further support the preferential reactivity of the redox-dependent anti-La mAbs 2F9, 312B, 32A and 27E to the reduced form of La protein, their reactivity to untreated wild-type La protein (Figure 9C, lanes 1), the triple cysteine mutant (Figure 9C, lanes 2), the mono cysteine mutant (Figure 9C, lanes 3), and the double cysteine mutant (Figure 9C, lanes 4) were analyzed by SDS–PAGE and immunoblotting. The data obtained for the anti-La mAb 312B are shown in Figure 9C. As already shown for the anti-La mAb SW5 (Figure 7C, upper blots) or the anti-La mAb 7B6 (Figure 7C, lower blots), the protein samples for SDS–PAGE were either prepared in the absence of ß-mercaptoethanol (Figure 9C, untreated) or in the absence of ß-mercaptoethanol, but treated with CuSO_4_ (2 mM) prior to sample preparation (Figure 9C, oxidized), or in the presence of ß-mercaptoethanol (Figure 9C, reduced). In all protein samples prepared under reducing conditions, only the monomeric La protein is detected with the anti-La mAb 312B (Figure 9C, reduced, lanes 1 to 4). In contrast to the anti-La mAbs SW5 and 7B6 (Figure 7C), the anti-La mAb 312B failed to react with the high molecular weight La-related protein bands but reacted only with the monomeric La protein (Figure 9C, untreated). The reactivity of wild-type La slightly reduced (Figure 9C, untreated, lane 1) compared to, e.g., the triple cysteine mutant (Figure 9C, untreated, lane 2). Moreover, after oxidation with CuSO_4_ wild-type La protein completely lost the reactivity to the anti-La mAb 312B (Figure 9C, oxidized, lane 1). In contrast, oxidation of the triple cysteine mutant has no more effect on the reactivity of the anti-La mAb 312B (Figure 9C, oxidized, lane 2). Similarly, the reactivity of anti-La mAb 312B to 50 kDa La protein is no more influenced by oxidation in the case of the mono cysteine mutant (Figure 9C, oxidized, lane 3). In contrast, oxidation of the double cysteine mutant almost completely abolishes the recognition by the anti-La mAb 312B (Figure 9C, oxidized, lane 4). Thus, with respect to oxidation, the mono cysteine mutant behaves like the triple cysteine mutant, while the double cysteine mutant behaves like wild-type La protein. Consequently, oxidation of the N-terminal cysteine residue Cys18 destroys the reactivity of the anti-La mAb 312B, while oxidation of the cysteine residues Cys232 and Cys245 does not affect the reactivity of the anti-La mAb 312B. The loss of reactivity caused by oxidation means that the cysteine residue Cys18 can form disulfide bridges with another La molecule under oxidative conditions.

A closer view of the data presented in Figure 9C (oxidized) still raises some further questions. Looking at the oxidized mono cysteine mutant (Figure 9C, oxidized, lane 3), two additional anti-La reactive protein bands were detected related to a dimer and even higher protein aggregates. Thus, the lack of the cysteine residue Cys18 in the La motif also confers the resistance against redox-dependent reactivity of the anti-La mAb 312B to these dimers and oligomers. In addition, these data show that dimers/oligomers cannot only be formed and stabilized by oxidation/disulfide formation via the N-terminal cysteine residue Cys18 in the La motif but also via the two cysteine residues Cys232 and Cys245 in the RRM2 domain. This interpretation is further supported when looking at Figure 9C, oxidized, lane 4. Although oxidation of the mono cysteine mutant destroys almost completely its reactivity to the anti-La mAb 312B, a weak reaction of the anti-La mAb 312B is still visible in Figure 9C, oxidized, lane 4. Obviously, besides the monomeric also a dimeric form exists in this oxidized La mutant preparation (Figure 9C, oxidized, lane 4). Interestingly, only a dimer but not higher molecular weight oligomers were detected. Thus, oxidation via the cysteine residue Cys18 may only lead to dimers. According to our data, the formation of a disulfide bridge between two La molecules via the cysteine residue Cys18 in the La motif can occur independently of the cysteine residues in the RRM2 domain. However, the formation of oligomers requires more than one cysteine residue in one La protein molecule.

In order to further support the redox sensitivity of the cysteine residue Cys18 in the La motif, an additional mutant was prepared as schematically shown in Figure 9DI. For this purpose, we started from the C-terminally truncated fragment LaN, which consists of the La motif and the RRM1 domain but lacks the RRM2 domain and thereby the cysteine residues Cys232 and Cys245. In case cysteine residue Cys18 contributes to the redox-dependent sensitivity of the 312B epitope, we expected that mutation of the cysteine residue Cys18 to alanine should also abolish the redox sensitivity of this fragment, supporting that the redox-dependent reaction of the La motif can occur independently of the redox reaction in the RRM2 domain. As expected, the anti-La mAbs SW5 and 5B9 react with the LaN fragment. Replacing the cysteine residue Cys18 (Figure 9DII–IV, gray bars) with alanine (Figure 9DII–IV, black bars) does not affect the reactivity of these anti-La mAbs (Figure 9DII, SW5, 5B9). Neither oxidation (Figure 9DIII, SW5, 5B9) nor reduction (Figure 9DIV, SW5, 5B9) remarkably influences their anti-La reactivities. In contrast, oxidation of the LaN fragment almost completely destroys the reactivity of the anti-La mAb 312B to wild-type LaN (Figure 9DIII, gray bars), while the reactivity to the Cys18 mutant is not affected by oxidation. Under reducing conditions, however, both the wild-type and the Cys18 mutant LaN fragment react equally well with the anti-La mAb 312B (Figure 9DIV, 312B). Obviously, a major portion of wild-type LaN is already oxidized in the untreated fraction (Figure 9DII, 312B, gray bar), which explains the limited reactivity to the anti-La mAb 312B with untreated LaN. The same results as for the anti-La mAb 312B were obtained for all other anti-La mAbs directed to the La motif, including 2F9, 32A, and 27E (data not shown).

The redox-dependency of these antibodies also becomes obvious from the apparent K_D_ values as estimated by ELISA for the oxidized (Figure 10, wild-type La, triangle) and reduced form (Figure 10, triple cysteine mutant, square) of La protein. While we estimated no dramatic difference between the K_D_ values to the oxidized and reduced form of La protein for the anti-La mAbs SW5, 5B9 and 7B6, there is a dramatic difference for the anti-La mAb 312B. We estimated for all mAbs K_D_ values around 10^−10^ M to the reduced form of La protein and similar K_D_ values to the oxidized form of La protein for the anti-La mAbs SW5, 5B9 and 7B6. However, the K_D_ value of the anti-La mAb 312B to the oxidized form of La protein is about 10-fold lower (around 10^−9^ M). According to these data, it should be possible to compensate for the dramatically reduced binding of the anti-La mAb 312B to the oxidized La form by increasing the concentration of oxidized La protein about 10-fold.

In summary, these data tell us that not only the two cysteine residues Cys232 and Cys245 of La protein are sensitive to oxidation as previously shown by their use in the ALARM NMR assay and supported by our data obtained for the anti-La mAb 7B6, but all three cysteine residues present in La protein are sensitive to oxidation, including the cysteine residue Cys18 in the La motif. The dimers and oligomers formed by oxidation are stabilized by disulfide bridges. Oxidation of the cysteine residue Cys18 in the La motif dramatically reduces the reactivity of the anti-La mAb 312B, which is in good agreement with the epitope mapping data. According to these data, the anti-La mAb 312B and all the other anti-La mAbs directed to the La motif (2F9, 32A, and 27E) recognize a conformational epitope in the La motif, which is sensitive to oxidation. Oxidation via the cysteine residue Cys18 can occur independently of the two cysteine residues Cys232 and Cys245 in the RRM2 domain of La protein.

### 2.8. Evidence for Redox Dependent Conformational Changes of La Protein

Hence, so far, our data point to redox-dependent conformational changes of the structure of La protein. We challenged this question by application of biophysical analyses. Circular dichroism (CD) spectra can be used to estimate the percentage of helices and ß sheets present in the secondary structure of a protein. If the cysteine residues in La protein are somehow relevant for its structure and sensitivity to oxidation, we expected that oxidation of wild-type La protein affects the secondary structure of wild-type La protein and thereby resulting in different content of helices and/or ß sheets, while the secondary structure of the triple cysteine mutant should be more or less resistant against oxidative conditions. Moreover, CD analysis of the mono- and double cysteine mutant should give us additional information, which cysteine residue(s) are involved in redox-dependent structural changes. Therefore, we determined the CD spectra of wild-type La protein (Figure 11A, La wild-type), the mono (Figure 11A, La_C_18_A), double (Figure 11A, La_C_232_A_C_245_A), and triple cysteine mutant (Figure 11A, La_C_18_A_C_232_A_C_245_A), including under oxidative conditions (Figure 11A, solid line (untreated), dashed line (oxidized by the presence of CuSO_4_). The CD spectra were measured between 200 and 260 nm. The resulting CD spectra support the sensitivity of La protein to oxidation. All cysteine residues contribute to this sensitivity: La protein is sensitive to oxidation as long as a single cysteine residue remains in the primary sequence (Figure 11A, La wild-type, mono cysteine mutant La_C_18_A, double cysteine mutant La_C_232_A_C_245_A). Only after replacement of all three cysteine residues, La protein becomes resistant to oxidative conditions (Figure 11A, triple cysteine mutant La_C_18_A_C_232_A_C_245_A). This becomes even more evident when we calculated and compared the percentage of helicity of the respective untreated versus oxidized La protein. The content of helices and ß sheets was calculated according to Sreerema et al. [52]. The data are summarized in Table 1. Oxidation of wild-type La protein causes a drop of helicity from about 55% to 29%, meaning La protein loses around 45% of its helical content during oxidation. In contrast, oxidation of the triple cysteine mutant does not change dramatically the helicity (untreated: 44%, oxidized 42%). In the case of the mono cysteine mutant, we found even a slight increase of helicity from 50% to 58%. For the double cysteine mutant, we calculated a drop of helicity from 63% to 47%. According to these data, the N-terminal cysteine residue Cys18 contributes the most to the oxidation-dependent loss of helicity.

### 2.9. According to the Melting Temperature, the Reduced Form of La Protein Has the Highest Stability

Next, we wanted to learn whether the secondary structure of La protein is more stable under reducing or oxidative conditions. In order to answer this question, we determined the melting temperature of wild-type La protein and the triple cysteine mutant of La protein in the absence and presence of CuSO_4_. For this purpose, we estimated and blotted the ellipticity measured by CD in a temperature range between 10 °C and 80 °C. As shown in Figure 11B, we see no major difference between the melting curve of the triple cysteine mutant in the absence (Figure 11B, La_C_18_A_C_232_A_C_245_A, red line) or presence (Figure 11B, La_C_18_A_C_232_A_C_245_A, orange line) of CuSO_4_. In contrast, we found a major difference between the melting curve for wild-type La protein in the absence (Figure 11B, La wildtype, blue line) or presence (Figure 11B, La wild-type, purple line) of CuSO_4_. Moreover, the slope of the curve for wildtype La protein in the presence of CuSO_4_ is clearly higher than in the absence. Based on these data, we also estimated the T_m_ values. The T_m_ for the triple cysteine mutant was estimated with 43 °C in the absence and 34 °C in the presence of CuSO_4_. The T_m_ for wild-type La protein was estimated with 42 °C in the absence and 28 °C in the presence of CuSO_4_. According to these data, wild-type La protein denatures faster under oxidative conditions. Replacing the cysteine residues with alanine residues stabilizes the secondary structure of La protein and results in resistance against changes of the secondary structure caused by oxidation.

In summary, these data support the conclusions: (i) All the cysteine residues in La protein contribute to the redox sensitivity of La protein and (ii) from a thermodynamic point of view, the conformation of the reduced form of La protein is more stable.

### 2.10. Redox Dependent Anti-La Antibodies Are Present in Sera of Autoimmune Patients

A first indication that redox-dependent changes also occur in vivo and contribute to the autoimmune response in patients would be the presence of Abs in anti-La sera of autoimmune patients reacting to La protein in a redox-dependent manner. Therefore, we estimated anti-La patient sera for a differential reactivity to oxidized and reduced La protein. Sixty-four anti-La reactive sera were tested. As shown in Figure 12, about half of the sera preferentially reacted with the oxidized and vice versa with the reduced form of La protein. These data show that redox-dependent epitopes also play a role in the anti-La response of autoimmune patients.

## 3. Discussion

Since the description of the autoantigen La/SS-B at the beginning of the 1970ties we, like many other groups, tried to elicit mAbs against it. One of these mAbs is the anti-La mAb 4B6 and its subclone 7B6 [46]. Over the years, the anti-La mAb was shared with many international labs. While the mAb reliably worked in ELISA and SDS–PAGE and immunoblotting studies, in few cases, we were told that the mAb completely fails in IF microscopy. The answer to this problem came by chance. Commonly, we store the methanol for fixation in plastic bottles in the fridge (see also Figure 2). For one experiment, a series of coverslips were prepared. In this case, the amount of methanol was not sufficient; therefore, the remaining coverslips were fixed with methanol of analytical grade, which was taken from a freshly opened dark brown glass bottle. As all the coverslips fixed with the methanol stored in the plastic bottle worked well, while all the coverslips failed, which were fixed using the methanol stored in the brown glass bottle, we speculated that perhaps oxygen radicals or peroxides might have been formed during the storage in the plastic flask, which should be absent in the analytical grade methanol. Indeed, when coverslips, which failed in the first staining round, were shortly rinsed with PBS, to which hydrogen peroxide was added, then the staining was nicely restored. From these data, we concluded that the accessibility of the epitope recognized by the anti-La mAb 7B6 depends on oxidative conditions. Consequently, La protein should be somehow sensitive to oxidation. Moreover, we expected that structural changes might occur in a redox-dependent manner, which could affect the accessibility of the epitope recognized by the anti-La mAb 7B6.

Such redox-dependent changes were not unlikely to occur. For the toxicological evaluation of thiol-reactive compounds, Huth et al. described a nuclear magnetic resonance (NMR)-based assay termed ALARM NMR [21,22,23]. While searching for a protein substrate for such an assay, the authors found that a portion of La protein (aa293-348) is most suitable for this purpose. The ALARM NMR assay was shown to be based on the two cysteine residues Cys232 and Cys245, which are present in the RRM2 domain of La protein.

Besides these two cysteine residues, La protein contains an additional cysteine residue (Cys18), which is located in the first helical domain of the La motif. Consequently, cysteine Cys18 is not present in the La fragment used for ALARM NMR. According to our data, not only the two cysteine residues Cys232 and Cys245 in the RRM2, but all the three cysteine residues contribute to the sensitivity of La protein to oxidation.

Epitope mapping for the anti-La mAb 7B6 shows that the mAb is directed to the α3 helix in the RRM2. Therefore, the cysteine residues (Cys232 and Cys245) locate upstream of the 7B6 epitope region and are not part of the 7B6 epitope. When fused to EGFP, the epitope is no more cryptic and accessible, including for IF microscopy without an oxidative treatment. Moreover, we successfully used the 7B6 epitope sequence as a protein tag in a series of studies [40,53,54,55,56,57,58,59,60,61,62,63,64,65,66,67,68,69,70,71,72]. Therefore, the 7B6 epitope itself is not sensitive to oxidation. Consequently, allosteric conformational changes may influence the accessibility of the epitope for the anti-La mAb 7B6.

The identified epitope sequence is part of a previously identified dimerization domain [14]. In this study, dimerization of La protein was reported to be important for its proposed cytoplasmic function in Cap-independent translation [14]. However, structural analyses argue against the dimerization of La protein [15]. Until now, only the structure of three fragments of La protein was obtained (La motif, RRM1, RRM2, see also Figure 1). The structure of full-length La protein is not available. It is noteworthy to mention that the structural studies were performed in the presence of reducing agents. This includes an ultracentrifugation analysis, which was performed under reducing conditions and, from which it was concluded that La protein is not able to form dimers [15]. In agreement with these studies, we see that La protein is monomeric under reducing conditions. However, our extended analyses support the sensitivity of La protein to oxidation. The sensitivity to oxidation may also be the reason why until now, the structure of La protein has not been fully solved. Different forms, including a three thiol form and various oxidative forms, including dimers, oligomers and polymers, may be formed, which of course, make a structural analysis quite challenging.

According to the here presented data, not only the La fragment used in the ALARM NMR but even wild-type La protein can undergo redox-dependent conformational changes. Moreover, all cysteine residues are sensitive to oxidation. The reduced form (three thiol form) of La protein seems to be the most stable form, and La protein is monomeric in the reduced form. Oxidative conditions allow La protein to flip in another conformation, which is less stable but can be stabilized by the formation of disulfide bridges. Thereby, the dimerization domain becomes available, allowing the La protein to form protein–protein interactions. These protein–protein interactions may occur within and between La molecules, but it may also lead to the formation of heterodimers with other proteins. These interactions may depend on the redox conditions existing in the respective cell or tissue.

Already in 1988, we described a nucleocytoplasmic shuttling of La protein, e.g., after UV irradiation and virus infections [34,35,36,37,38]. Since then, similar data were published by other independent groups [73,74,75,76]. All the observed conditions lead to such a cytoplasmic translocation of La protein cause intracellular oxidative stress. In order to control and repair DNA damage, cells go through a cell cycle. During interphase, translation of mRNAs into protein commonly occurs in a Cap-dependent manner. Protein synthesis, however, switches to Cap-independent translation during cell division. According to a series of studies, La protein plays a role in translational control, including in Cap-independent translation. Bearing in mind that both the 7B6 epitope and the NRE is part of the dimerization domain, La protein may interact with a nuclear binding partner under reducing conditions, which retains La protein in the nucleus. Oxidative stress conditions destabilize this interaction. As a consequence, La protein can shuttle to the cytoplasmic compartment helping to translate mRNAs containing an IRES element. This interpretation could explain why the 7B6 epitope is cryptic under physiological reducing conditions in the nucleus but becomes accessible under oxidative stress conditions. In an oxidative environment, the oxidative conformation of La protein could be stabilized by disulfide bridges leading to dimers, oligomers and even larger protein aggregates.

In a previous study, the dimerization domain was identified in the RRM2 domain of La protein. In contrast to these studies, we found that protein–protein interactions are not only mediated via the dimerization domain in the RRM2 of La protein, but also the La motif contributes to the formation of dimers. Moreover, not only the cysteine residues (Cys232 and Cys245) but also the cysteine residue Cys18 in the La motif is highly sensitive to oxidation. Oxidation of the cysteine residue Cys18 leads to conformational changes of La protein. This became evident from the redox-dependent reactivities of the four anti-La mAbs (312B, 2F9, 32A, 27E) directed to the La motif. The data of the CD studies support these findings. For wild-type La protein and all La mutants, which contain at least a single cysteine residue, we can see redox-dependent changes of the secondary structure, especially with respect to the helicity of La protein. Interestingly, the cysteine residue Cys18 is part of the first N-terminal α-helix in La protein. Thus, oxidation of the cysteine residue Cys18 may influence this helix and thereby the conformation of the La motif. Similarly, oxidation of the cysteine residues (Cys232 and Cys245) could also change the structure of the RRM2 domain and thereby improve the accessibility of the α3-helix in the RRM2 domain and thereby the epitope recognized by the anti-La mAb 7B6. This becomes likely when looking at the structure of the RRM2 domain. Although the cysteine residues (Cys232 and Cys245) are not directly part of the epitope recognized by the anti-La mAb 7B6, the cysteine residue Cys232 locates in the vicinity to the α3-helix recognized by the anti-La mAb 7B6. In the reduced form of the RRM2 domain, the cysteine residue Cys232 locates upstream of a ß sheet followed by a flexible aa sequence, while Cys245 locates downstream of these elements [15,16,17,18,19]. For the formation of a disulfide bridge, as was shown to occur by NMR analysis [21,22,23], the two cysteine residues must be brought in close vicinity. Looking at the predicted structure of the RRM2 domain, there is certainly no doubt that this must lead to structural changes in the protein.

The claimed protein–protein interaction regulated by redox-dependent conformational changes must not necessarily be limited between the N- and C-terminal domain of the same La protein molecule or between different La protein molecules. In this context, it is noteworthy to mention that the cysteine residue Cys18 in the N-terminal La motif of bona fide La protein also exists in the La motif of LARP4, which localizes and functions as poly(A)-binding protein in the cytoplasm. It will, therefore, be of interest to test whether bona fide La protein can also form heterodimers with the La motif of LARP proteins in a redox-dependent manner. Such protein interactions could be a key step in switching Cap-dependent to Cap-independent translation and translational control.

Finally, one may ask the question of whether or not such redox reactions occur at all in vivo. The best indication is the finding that sera of patients contain anti-La Abs that preferentially react with either the reduced or oxidized form of La protein. Previous studies have shown that the management of oxidative stress is less effective in Lupus patients [7,8,9,10,11,12,13,14,15,16,17,18,19,20,21,22,23,24,25,26,27,28,29,30,31,32,33,34,35,36,37,38,39,40,45,46,47,48,49,50,51,52,53,54,55,56,57,58,59,60,61,62,63,64,65,66,67,68,69,70,71,72,73,74,75,76,77,78,79,80,81,82,83,84]. Moreover, autoantigens undergo redox-dependent modifications that can trigger autoimmune responses [82]. Altered redox-dependent conformational changes in autoimmune patients may tip the balance in favor of an oxidized conformation whereby cryptic epitopes may become accessible and facilitate such redox-dependent modifications.

According to our data, we would expect that La protein becomes oxidized prior to the export to the cytoplasm. Consequently, an anti-La mAb preferentially recognizing the reduced nuclear form might not be able to detect the oxidized form and vice versa leading to controversial data. The number of posttranslational modifications, e.g., phosphorylation, may further challenge the interpretation of Ab-based data. As also shown here, the murine epitope sequence recognized by the anti-La mAb 7B6 is quantitatively modified, most likely phosphorylated in a eukaryotic cell, while the human epitope is only partially modified. Consequently, the anti-La mAb 7B6 works only for La protein exposed to oxidative conditions and still shows only the localization of the non-posttranslationally modified portion of La protein in human cells. It cannot be used for mouse cells. A similar problem may exist for the commonly used anti-La mAb SW5. Like the anti-La mAb 7B6, it reacts with both recombinantly in *E. coli* expressed human and mouse La protein [45,48,49,50]. However, it fails to recognize La protein expressed in mouse cells [45]. Again, this points to a posttranslational modification of the epitope recognized by the anti-La mAb SW5, which is complete in mouse cells. This posttranslational modification is yet unclear and whether or not a portion of human La protein is modified in the same manner remains open, thus limiting the information that can be drawn from experiments using just this antibody. Until today, many controversial data have been reported for the localization and function of La protein. These controversial data may simply be caused by the fact that only a subpopulation of La protein was detected or followed in the respective study. For example, comparing the ELISA and immunoblotting data obtained with the redox-dependent anti-La mAbs directed to the La motif (312B, 2F9, 32A, and 27E) with the data obtained with Abs directed to the RRM1 domain (SW5) or the linker between the La motif and the RRM1 (5B9) tells us that La protein recombinantly expressed in *E. coli* is a mixture of oxidized and reduced portions as well as monomers, dimers and oligo-/polymers. Bearing in mind the high sensitivity of La protein to oxidation, the ratios of the different forms in such a mixture varies. It depends on the expression, preparation of the extract, and purification procedure. It can also be affected during storage and thawing. Moreover, similar limitations exist if structural analyses are performed using fragments and IF microscopy, including when using polyclonal anti-La sera representing different mixtures of anti-La Abs. The use of La EGFP fusion proteins does also not solve this problem as proteolytic cleavages may occur, influencing the interpretation. Thus, as performed in this study, double IF studies using anti-La mAbs in combination with EGFP La fusion proteins have the advantage to identify such posttranslational modifications or alterations.

In summary, here we presented evidence that La protein is sensitive to oxidation. Oxidation can lead to the formation of dimers, and higher oligomers, which were not detected in previous structural studies because (i) these studies were performed under reducing conditions, thus favoring the formation of the reduced monomeric form of La protein, and (ii) until now tools were not available to experimentally differentiate between different (redox-dependent) conformations of La protein.

## 4. Materials and Methods

In general, if not noted otherwise, all experiments were performed in triplicates. Statistical analysis was determined using GraphPad Prism software 7.0 (GraphPad Software Inc.). All error bars are represented either as the standard error of the mean (SEM) or standard deviation (SD). If not indicated, SD was below 5%.

### 4.1. Recombinant Human La Protein Expression and Characterization

Recombinant human La protein, La-related deletion mutants (La_1-94_, La_1-100_, La_1-104_, La_1-112_, La_1-126_, La_1-192_, La_1-245_, La_1-310_, La_1-344_, La_1-371_, La_159-408_, La_194-408_, La_239-408_, La_278-408_, La_318-408_, La_369-397_, La_369-408_, La_295-408_, La_346-408_, La_369-397_, La_369-408_, La_376-408_, La_5-100_, La_10-100_, La_15-100_, La_20-100_, La_86-100_, La_90-104_, La_94-104_, La_82-112_, La_107-200_, La_120-245_, and the mono (La_C_18_A), double (La_C_232_A_C_245_A) and triple cysteine mutant (La_C_18_A_La_C_232_A_C_245_A) were cloned into the vector pET28b for expression of His-tagged proteins in *E. coli* and in the vector pEGFP-C2 with or without a C-terminal NLS. The respective La reading frames were amplified by PCR. Proteins expressed in *E. coli* were purified using nickel affinity chromatography as described previously [85]. The La EGFP fusion constructs were used for transfection of human HeLa or mouse 3T3 cells. Total extracts were analyzed by SDS–PAGE and immunoblotting or by IF microscopy. Prior to epifluorescence microscopy, cells were fixed with methanol (MeOH). As described in the result section, we used methanol of different quality, including methanol of analytical grade, which was stored in a brown glass bottle or methanol, which was stored in a plastic bottle. In the case fixed cells could not be stained by the anti-La mAb 7B6, the coverslips were washed for 5 min in PBS containing 30 μM H_2_O_2_. Fixed cells were stained with hybridoma cell culture supernatants and analyzed by IF microscopy as described previously [86]. Alternatively, total cell extracts were prepared and used for SDS–PAGE and immunoblotting [85,87].

SDS gel and immunoblot image acquisition were done with the ChemiDoc^TM^ MP Imaging System (Bio-Rad Laboratories). Data were subsequently analyzed via ImageLab^TM^-Software, Version 5.2.1 (Bio-Rad Laboratories).

For 3D-modeling of the La-motif, RRM1 and RRM2 of La protein, we used the program MOLMOL and the published data (PDB: 1S7A La-motif, 1S9A RRM1, 1OWX RRM2) [15,18].

### 4.2. Anti-La mAbs

In the present studies we used for ELISA, IF microscopy, and immunoblotting cell culture supernatants of hybridomas secreting the previously described anti-La mAbs SW5 [49,50], 7B6 [45,46], 5B9 [51], 312B, 2F9, 32A, 27E [45].

### 4.3. Redox ELISA

For ELISA, we used the respective recombinant human La protein or protein fragment, which was expressed in *E. coli* and isolated by nickel affinity chromatography. The respective La protein samples were coated to ELISA plates, and ELISA was performed using the respective anti-La mAb or patient anti-La sera as described previously [88]. The ethics committee of the OMRF approved the use of the patient’s sera (IRB approval #01-06 from 21 August 2002). Protein samples were coated either as obtained after purification (untreated) or in coating buffer containing 2 mM of CuSO_4_ (oxidized) or 5 mM DTT (reduced).

### 4.4. Estimation of CD Spectra

CD spectra of wild-type La protein, the mono, double, and triple cysteine mutant, including under oxidative conditions, were measured between 200 and 260 nm. For the analysis, we used protein samples at a concentration of 0.2 mg/mL dissolved in PBS containing or lacking 90 µM CuSO_4_. The content of α helices, ß sheets, turns, and unstructured regions were calculated using the CDSSTR-algorithm (modification of variable selection method) according to Johnson [89,90,91] and the reference database 4 for protein analysis in the wavelength between 190 to 260 nm [52,92].

In order to estimate the melting temperature, thermal denaturing experiments were performed as follows: CD spectra were collected in the temperature range between 10 and 80 °C. Each step represents a difference of 2.5 °C. Data were evaluated according to Greenfield [93].

### 4.5. Surface Plasmon Resonance

Surface plasmon resonance measurements were performed and evaluated as described in detail [94].

## Figures and Tables

**Figure 1 ijms-22-03377-f001:**
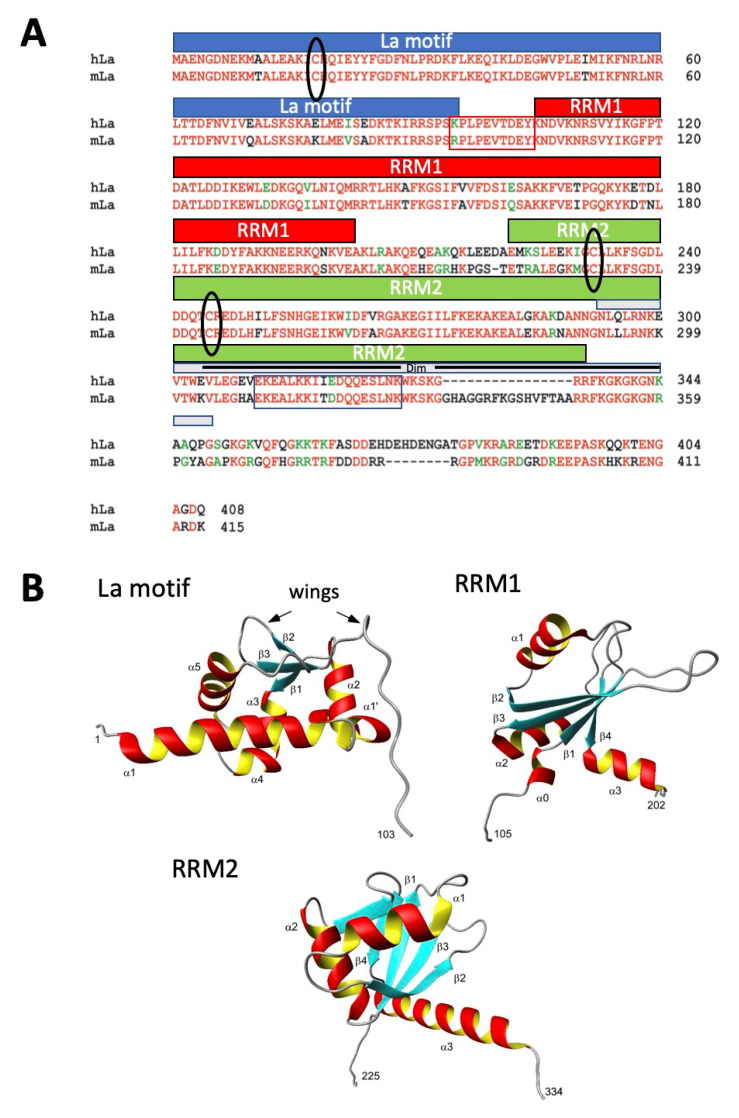
The La protein sequence and structure. (**A**) La protein is conserved during evolution [41,42]. For a detailed review of the sequence and structural comparisons, see also [17]. La proteins contain an N-terminal La motif (blue bar) followed by an RNA recognition motif (RRM1, red bar). The La motif contains an evolutionarily conserved cysteine residue (Cys18). La motif and RRM1 together form the La module (LaM) [17]. In previous studies, the LaM was termed LaN [41,42,43,44]. The sequence between the La motif and the RRM1 is unstructured but forms a helical structure in the case La protein binds to RNA [17]. The monoclonal anti-La antibody (anti-La mAb) 5B9 recognizes this linker (aa311-328 (KPLPEVTDEY, red box)) between the La motif and the RRM1 domain. Recently, mAbs were described by us being directed to conformational epitopes in the La motif (312B, 2F9, 32A, 27E) and RRM1 (22A, 24G7) [45]). Downstream of RRM1 La protein contains a second RRM (RRM2). The RRM2, together with the downstream sequence, represents the previously termed C-terminal La fragment LaC [41,42,43,44]. The RRM2 contains a dimerization domain (Dim, gray bar) [14]. Within this region, a nuclear retention element (NRE) [15] and a nucleolar localization signal (NoLS) [16] were identified. The epitope sequence (EKEALKKIIEDQQESLNK (aa311-328)) recognized by the anti-La mAb 7B6 is also part of this region (blue box, see also below). The data presented below show that aa304-344 is sufficient for dimerization (black line in the gray bar). The RRM2 domain contains the two cysteine residues (Cys232 and Cys245), playing a central role in the ALARM NMR assay [21,22,23]. (**B**) The structure of La protein is partially solved [17,18,19]. Structural data are available for the La motif, the RRM1 and RRM2 domain. The La motif contains two “wings”. The interdomain sequence, which contains the epitope recognized by the anti-La mAb 5B9, starts downstream of the second wing. The epitope of anti-La mAb 7B6 includes most of the α3-helix in the RRM2 domain.

**Figure 2 ijms-22-03377-f002:**
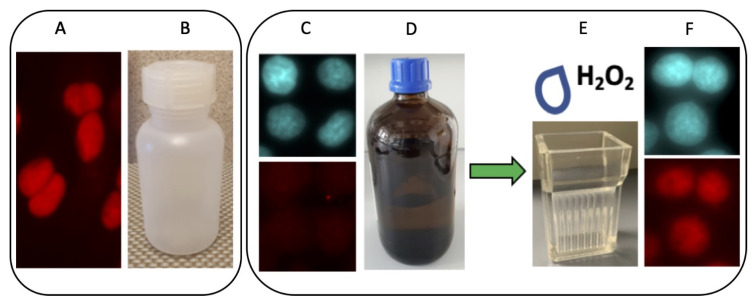
Accessibility of the epitope recognized by the anti-La mAb 7B6. Influence of fixation. (**A**) Cells were stained with the anti-La mAb 7B6. (**B**) The cells in (**A**) were fixed with methanol that was stored in a plastic bottle. (**C**) Cells were stained by the anti-La mAb 7B6 (lower panel). DAPI staining was performed to visualize the presence of cells (upper panel). Cells in (**C**) were fixed with methanol of analytical grade that was stored in a dark brown glass bottle (**D**). A single drop of hydrogen peroxide was added to the PBS in the washing chamber (**E**), and the staining procedure of the cells in (**C**) was repeated (**F**).

**Figure 3 ijms-22-03377-f003:**
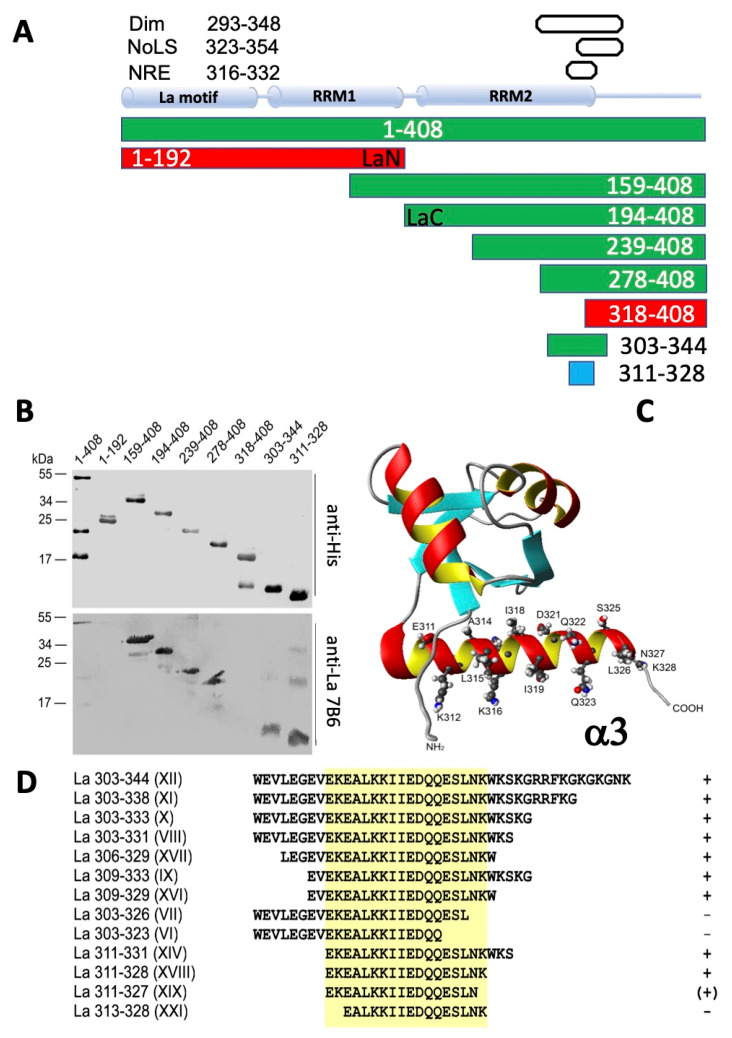
The epitope recognized by the anti-La mAb 7B6 is part of the predicted dimerization domain (Dim), nuclear retention element (NRE) and nucleolar localization signal (NoLS). (**A**) A series of deletion mutants, which were truncated from either the N- or the C-terminus or from both sites, were cloned as His-tagged fusion proteins, expressed in *E. coli* and isolated by nickel affinity chromatography. (**B**) The deletion mutants were tested against either anti-His Abs (upper panel) or the anti-La mAb 7B6 (lower panel). Results for selected deletion mutants are shown, including the identified epitope sequence aa311-328 (EKEALKKIIEDQQESLNK). (**C**) The epitope sequence consists of most of the α3-helix in the RRM2 plus 3 aa (LNK) of the unstructured region located C-terminally of the α3-helix (see also Figure 1). (**D**) Besides the deletion mutants shown in (**A**,**B**), a series of further fragments related to the epitope sequence recognized by the anti-La mAb 7B6 were prepared and tested, which finally helped us to locate the epitope recognized by the anti-La mAb 7B6 to the aa311-328.

**Figure 4 ijms-22-03377-f004:**
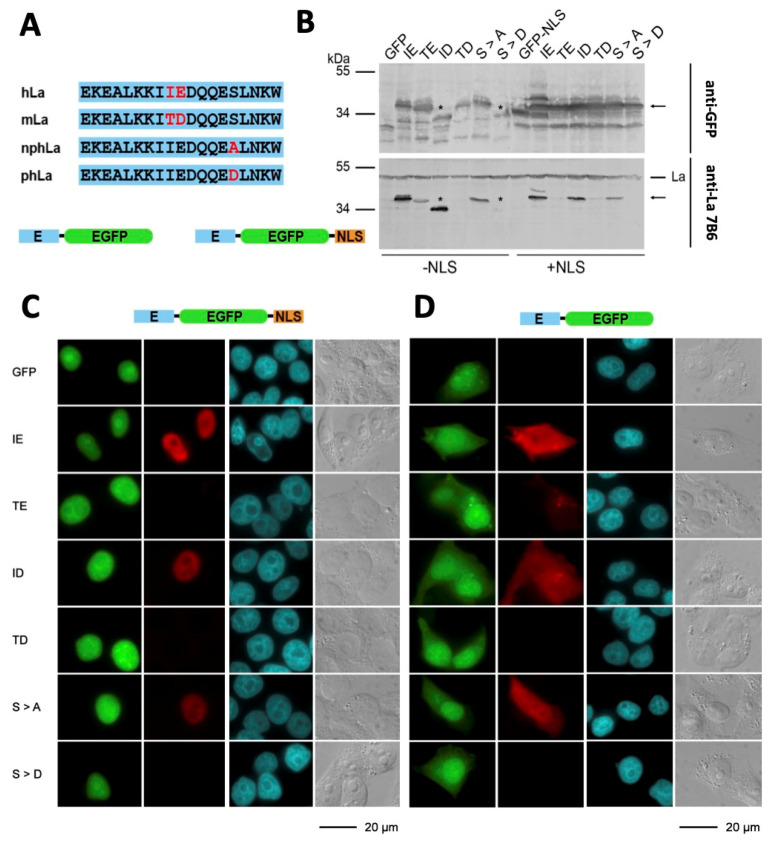
Posttranslational modifications in the epitope sequence of anti-La mAb 7B6. (**A**) The human (hLa) and the corresponding mouse (mLa) 7B6 epitope sequence differ with respect to the aa highlighted in red. The two aa IE in the human sequence is replaced by aa residues TD. Epitope (E) fusion proteins with an enhanced green fluorescent protein (EGFP) with (E-EGFP-NLS) or without a nuclear localization signal (NLS) (E-EGFP) were constructed. In addition, mutants were constructed in which either the aa isoleucine was replaced by threonine (TE) or the glutamate was replaced by aspartate (ID) (see **B**). The human epitope sequence was reported to undergo phosphorylation in serine 325 [48]. Therefore, we replaced serine 325 with alanine (S > A) or aspartate (S > D). After transfection, total extracts were prepared and analyzed by SDS–PAGE/immunoblotting (**B**). Alternatively, cells were fixed and analyzed by IF microscopy (**C**,**D**). (**B**) Cellular extracts were tested for their reactivity to anti-EGFP Abs ((**B**) anti-GFP, upper panel) or the anti-La mAb 7B6 ((**B**) anti-La 7B6, lower panel). Cells were manipulated to express either the (**C**) E-EGFP-NLS or (**D**) E-EGFP fusion protein and were analyzed by IF microscopy. Transfected cells were identified by the green fluorescence of EGFP (**C**,**D**). The reactivity of the anti-La mAb 7B6 was detected in the red channel (**C**,**D**). Non-transfected cells were identified besides transfected cells via DNA staining with DAPI (blue channel, **C**,**D**). * Due to a slightly different cloning procedure the linker upstream of the His-tag is smaller which explains the lower molecular weight of these fusion proteins.

**Figure 5 ijms-22-03377-f005:**
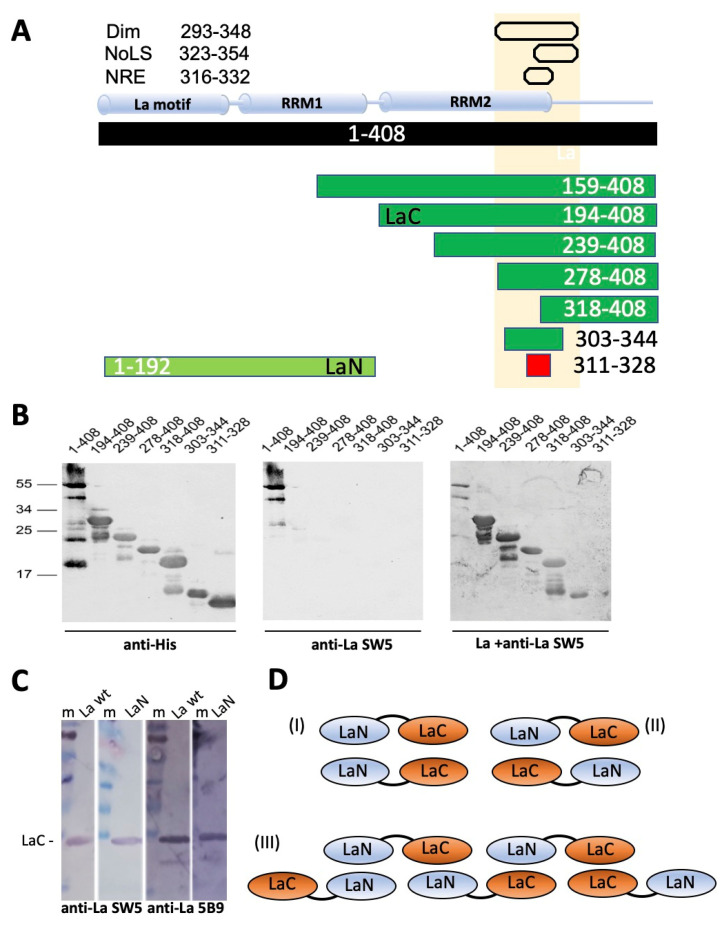
Dimerization of La protein. Craig et al. reported [14] that La protein contains a dimerization domain (Dim). Within the same region, an NRE [15] as well as a NoLS [16] was identified in independent studies. Structural data argue against a dimerization of La protein [15]. The epitope region recognized by the anti-La mAb 7B6 is also part of the predicted Dim domain and consists mainly of the α3-helical domain in the RRM2. (**A**) La fragments truncated either from the N- or C-terminus or from both sites, including the epitope region recognized by the anti-La mAb 7B6 (aa311-328), were expressed in *E. coli* as His-tagged proteins and purified by nickel affinity chromatography. (**B**) Far-Western blotting analysis was performed. All La-related fragments were separated by SDS–PAGE, and their presence was verified by immunoblotting using an anti-His Ab (**B**, anti-His). None of the fragments, but full-length wild-type La protein (**B**, 1–408) reacted with the anti-La mAb SW5 (**B**, anti-La SW5), which requires N- and C-terminal portions of the RRM1 domain for reactivity [49,50]. Incubation of the blotted La fragments with full-length La protein leads to a stable protein–protein interaction, which could be detected with the anti-La mAb SW5 (La +anti-La SW5). Using this modified Far-Western blotting assay, we identified the fragment aa303-334 as the smallest fragment, which is still able to interact with full-length La protein. (**C**) The LaN fragment (aa1-192) lacks the Dim domain. Still, after Far-Western blotting, binding of LaN to LaC was detected by both the anti-La mAb SW5 and 5B9. (**D**) (I) According to these data, besides the proposed head–head/tail-tail dimers [14] also head to tail dimers (II) should be possible and also a mixture of oligomers and even polymers may be formed (III).

**Figure 6 ijms-22-03377-f006:**
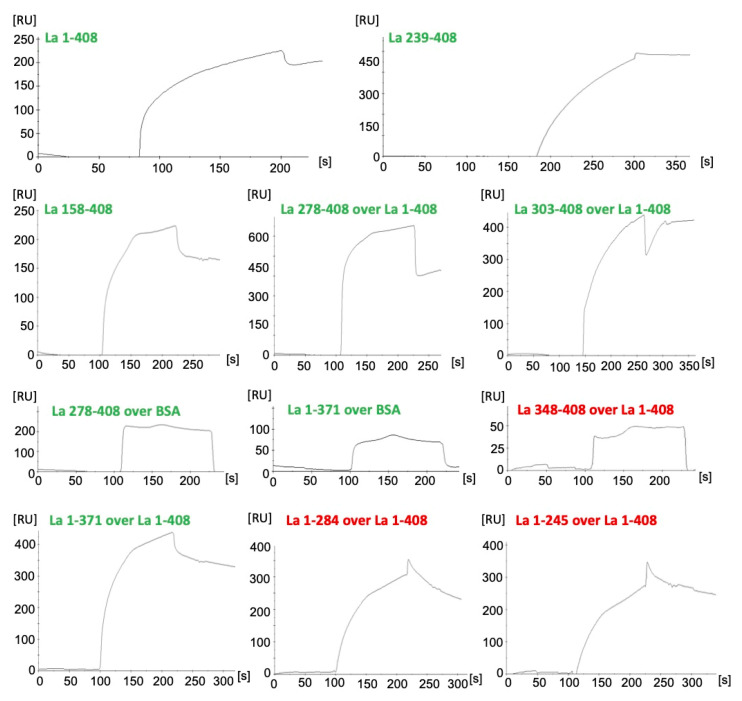
SPR studies to confirm La protein–protein interactions. Wildtype La protein was covalently linked to a sensor chip. Binding of full-length La protein (La 1-408) or N- or C-terminally truncated deletion mutants containing (highlighted in green) or lacking (highlighted in red) the identified Dim domain were tested for binding. As negative control served a sensor chip to which bovine serum albumin (BSA) was coupled.

**Figure 7 ijms-22-03377-f007:**
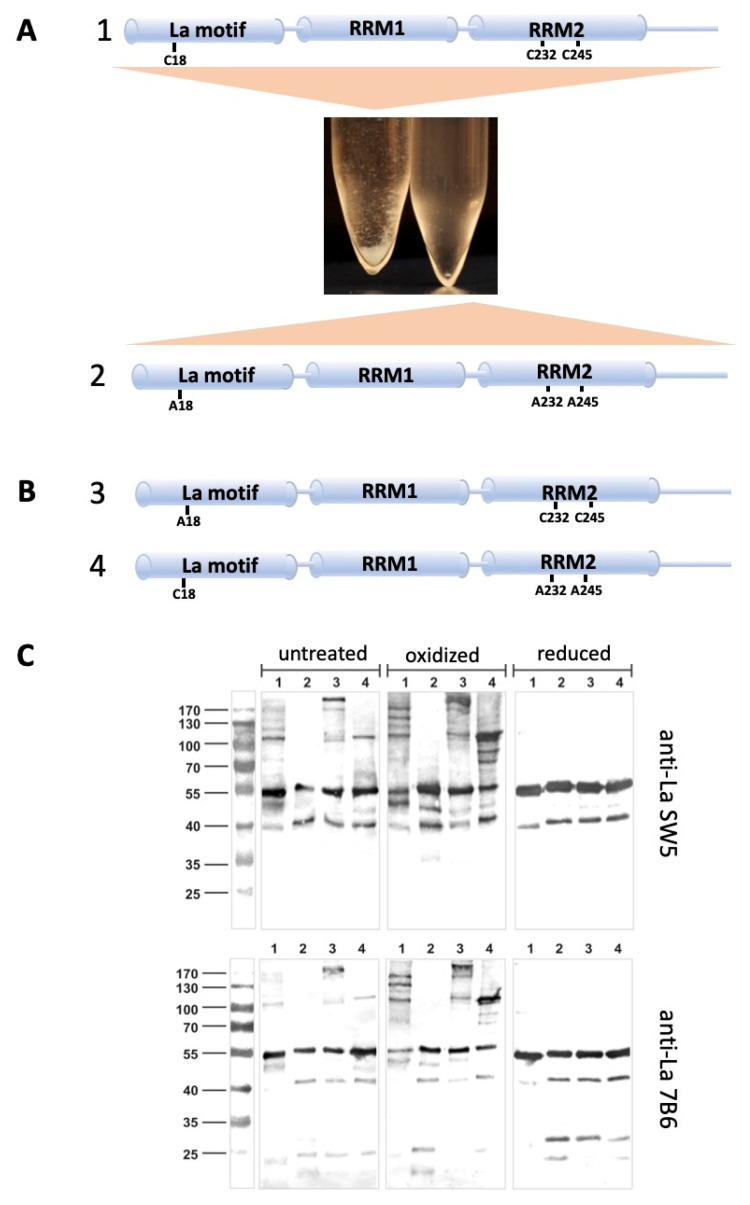
La protein forms dimers and higher oligomers in a redox-dependent manner. (**A**) As schematically drawn in (1), La protein contains three cysteine residues. One cysteine residue Cys18 (C18) in the La motif and two cysteine residues Cys232 (C232) and Cys245 (C245) in the RRM2 domain. When recombinant La protein is stored in the fridge or at room temperature for several hours, it tends to precipitate ((**A**, 1), left tube). Mutating all the three cysteine residues (C18, C232, C245) to alanine residues (A18, A232, A245) abolishes this precipitation ((**A**, 2), right tube). We assumed that the cysteine residues in La protein form disulfide bridges within or between La molecules, finally leading to insoluble oligomers/polymers. (**B**) In order to differentiate which of the cysteine residue(s) contribute to this oxidation-dependent polymerization, additional mutants were prepared in which (3) either the cysteine residue Cys18 (C18) in the La motif was mutated to alanine (A18, mono cysteine mutant) or (4) both cysteine residues Cys232 and Cys245 (C232, C245) were mutated to alanine (A232, A245, double cysteine mutant). (**C**) SDS–PAGE/immunoblotting data of untreated, oxidized and reduced samples of wild-type La protein (lanes 1), triple cysteine mutant (lanes 2), mono cysteine mutant (lanes 3), and double cysteine mutant (lanes 4). As “untreated” samples, we used the respective La proteins as isolated by nickel affinity chromatography. Prior to SDS–PAGE, the proteins were prepared in a sample buffer lacking reducing agents. Oxidized samples were obtained by incubation of the respective proteins in the presence of CuSO_4_. As for untreated samples, the proteins were prepared for SDS–PAGE in sample buffer lacking reducing agents. After separation by SDS–PAGE and transfer to blotting membranes, the respective samples were analyzed using either the anti-La mAb SW5 or anti-La mAb 7B6.

**Figure 8 ijms-22-03377-f008:**
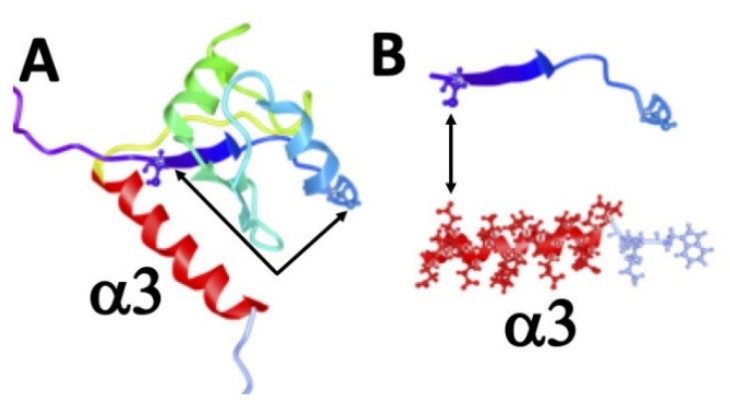
The α3-helix and the cysteine residues in the RRM2 domain**.** (**A**) The anti-La mAb 7B6 is directed to the α3-helix of the RRM2 domain. The accessibility of the epitope is sensitive to oxidation. The RRM2 contains two cysteine residues Cys232 and Cys245 (see arrows). (**B**) The cysteine residue Cys232 locates upstream of the ß-sheet1, while the cysteine residue Cys245 locates downstream of the ß-sheet in an unstructured region. In the RRM2 domain, the two cysteine residues are located at opposite sites (see arrows in (**A**,**B**)). In order to form a disulfide bridge, both cysteine residues must be positioned in close vicinity, which would require a refolding of the RRM2 domain.

**Figure 9 ijms-22-03377-f009:**
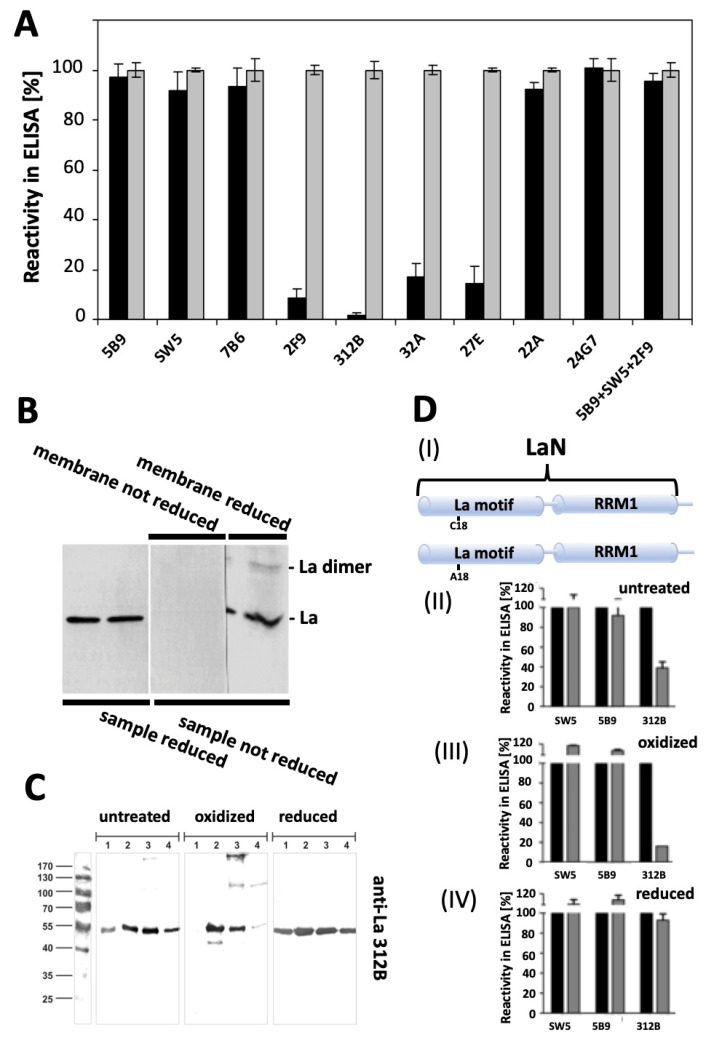
Identification of anti-La mAbs to the reduced form of La protein. (**A**) The reactivities of the anti-La mAbs 2F9, 312B, 32A, and 27E, which are directed against the La motif and the anti-La mAbs 22A, 24G7, which are directed against the RRM1 domain [45] were tested by ELISA against wild-type La (black bars) and the triple cysteine mutant (gray bars). In parallel, the reactivities were determined for the anti-La mAbs 5B9, 7B6, SW5. (**B**) SDS–PAGE/immunoblotting using the anti-La mAb 2F9. A recombinantly expressed wild-type La protein sample was treated prior to electrophoresis under reducing conditions (sample reduced). The same protein sample was not reduced prior to electrophoresis (sample not reduced, membrane not reduced). The latter blot was cut into two halves. One-half was rinsed in PBS under reducing conditions and tested again against the anti-La mAb 2F9 (sample not reduced, membrane reduced). The position of wild-type La and dimers of La are indicated. (**C**) SDS–PAGE/immunoblotting using the anti-La mAb 312B. Wild-type La protein (lanes 1), the triple cysteine mutant (lanes 2), the mono cysteine mutant (lanes 3) and the double cysteine mutant (lanes 4) were analyzed. La protein samples were used as obtained after nickel affinity chromatography (untreated). Alternatively, the protein samples were treated with CuSO_4_ prior to electrophoresis (oxidized). In both cases, the protein samples were not reduced prior to electrophoresis. As a third alternative, the protein samples were reduced prior to electrophoresis (reduced). (**D**) Redox sensitivity of the La motif. An ELISA was performed similarly to (**A**) using instead of wild-type La protein the C-terminally truncated La fragment LaN (gray bars). The LaN sequence contains only the cysteine residue Cys18. In parallel, we tested a mutant LaN fragment in which the cysteine residue Cys18 was mutated to alanine (black bars). The used La fragments are schematically shown in (I). The LaN fragments protein samples were used as obtained after nickel affinity chromatography (II, untreated) or treated with CuSO_4_ during coating of the ELISA plates (III) or reduced during coating (IV, reduced).

**Figure 10 ijms-22-03377-f010:**
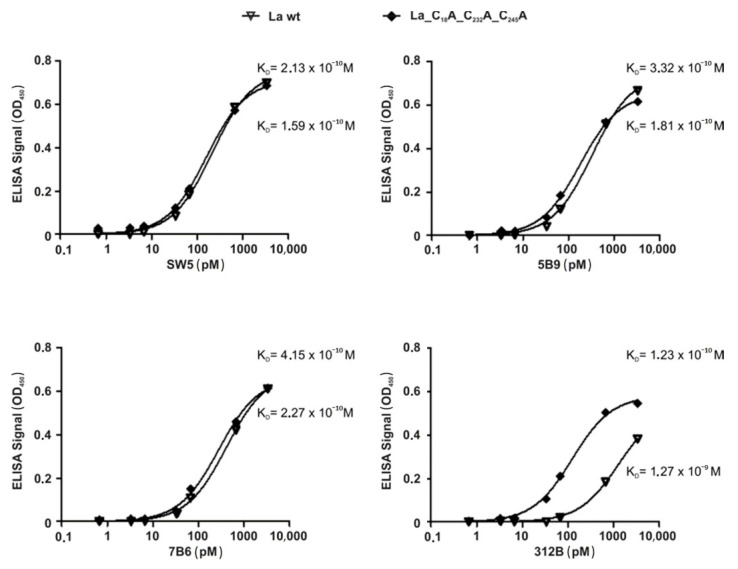
Comparison of apparent K_D_ values of anti-La mAbs to the reduced and oxidized form of La protein. Wildtype La protein (wt) and the triple cysteine mutant (La_C_18_A_C_232_A_C_245_A) were treated with CuSO_4,_ and the K_D_ values were determined by ELISA for the anti-La mAbs SW5, 5B9, 7B6, and 312B.

**Figure 11 ijms-22-03377-f011:**
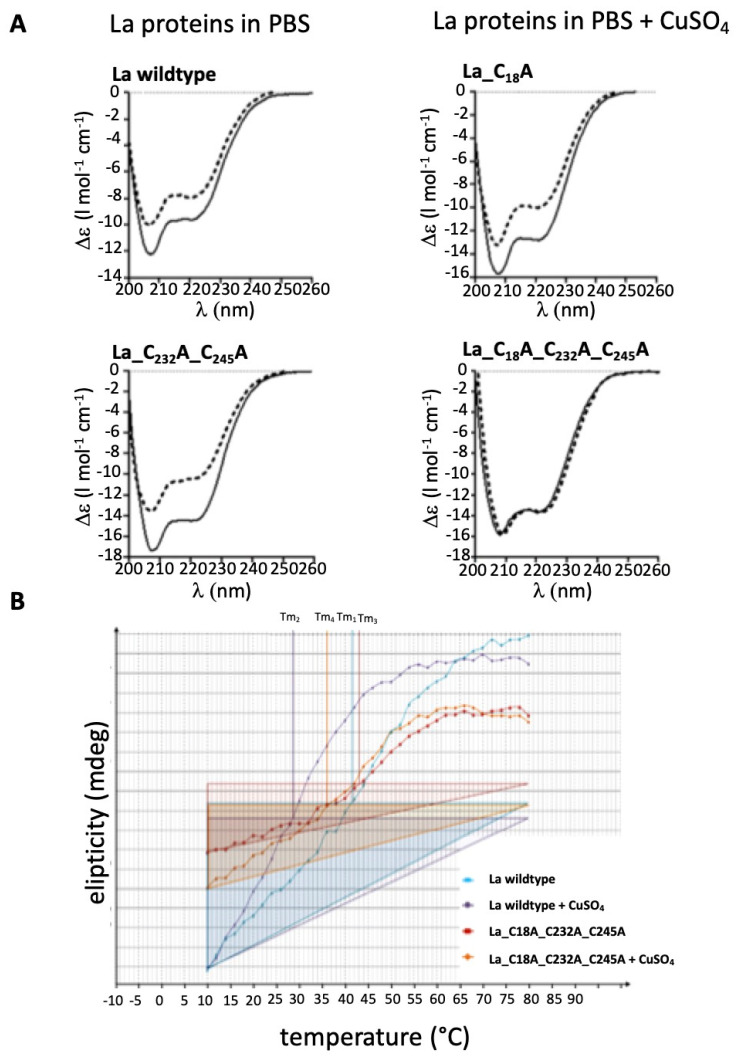
Circular dichroism (CD) analysis. (**A**) CD spectra were prepared for wild-type La protein (La wild-type), the mono (La_C_18_A), double (La_C_232_A_C_245_A), and triple cysteine (La_C_18_A_C_232_A_C_245_A) mutant, including under oxidative conditions. (**B**) Estimation of the melting temperature of La wild-type and the triple cysteine mutant (La_C_18_A_C_232_A_C_245_A) in the absence or presence of 90 μM CuSO_4_. The protein concentration of each sample was 0.2 mg/mL. CD spectra were collected in the temperature range between 10 °C and 80 °C. Each step represents a difference of 2.5 °C.

**Figure 12 ijms-22-03377-f012:**
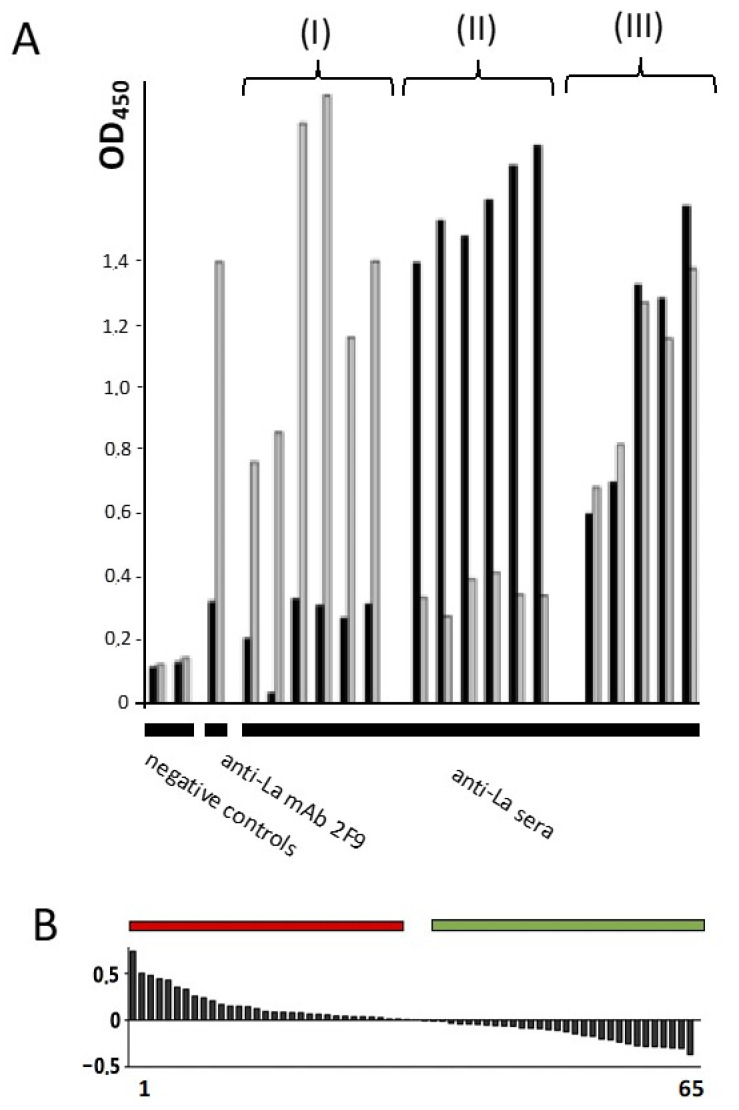
Comparison of the reactivity of patient’s anti-La sera against oxidized versus reduced La protein. Wild-type La protein was coated to ELISA plates. Prior to ELISA, La protein was either treated with H_2_O_2_ (black bars) or DTT (gray bars). Oxidoreduction was verified by the anti-La mAb 2F9. (**A**) Tested sera were grouped into three categories: (I) sera, which preferentially react with the reduced form of La protein, (II) sera, which preferentially react with the oxidized form of La protein, (III) sera, which do not show major differences in dependence on oxidoreduction. (**B**) Until now, 64 anti-La positive patient sera were tested for a differential reactivity to either the oxidized or reduced form of La protein. The sera were sorted for their preferential reactivity against either the oxidized (red bar) or reduced form of La protein (green bar). Each black bar represents one serum.

**Table 1 ijms-22-03377-t001:** Relative secondary structure contributions determined from CD-spectra (Figure 11) with dichroweb (http://dichroweb.cryst.bbk.ac.uk/html/home.shtml).

	*Wildtype La*		*La_C_18_A*		*La_C_232_A_C_245_A*		*La_C_18_A_C_232_A_C_245_A*	
helix 1	55	29	50	58	63	38	44	42
strand	16	26	21	17	13	23	15	17
turns	7	2	8	5	6	9	11	1
random	21	25	25	19	17	2	21	22
total	99	100	99	99	99	99	100	100

## Data Availability

The data presented in this study are available on request from the corresponding author.

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
