# Peer review of "Two Be or Not Two Be: The Nuclear Autoantigen La/SS-B Is Able to Form Dimers and Oligomers in a Redox Dependent Manner"

_ijms, 2021, doi:10.3390/ijms22073377_

Round 1

Reviewer 1 Report

The manuscript by Berndt et al provides biochemical evidence for the existence of oxidized form of La that can form dimers and that this form is immunogenic. Overall, the manuscript contains large body of data that support the conclusions reached by the authors.

Enclosed several comments for the authors:

  1. P3 – it is not clear how the authors came to hypothesize that the La protein will be sensitive to oxidation. The background about the cysteine residues does not provide the needed linkage.
  2. P3 section 2.1 – it is not clear what the authors mean by saying that by subcloning of 4B6 they created 7B6. Does it means that the 4B6 was not a purified clone? Did they subcloned the hybridomas and found that it was composed of several antibody-secreting clones that one of them was 7B6?
  3. P3 – if the binding of 7B6 is methanol-oxidation dependent, how was it characterized in the first place?
  4. Lines 200-202 – why it was chosen to use these two cell lines? And why two?
  5. Line 267 – 7B6 epitope? Instead of sequence?
  6. Line 666 – I believe that from ELISA one can determine apparent KD and not accurate KD.
  7. Figure 9 – was the data fitted? If yes, please mention how. In addition, what value was set as the Bmax? The Bmax value should be identical for each set of measurements. Currently, it seems that it was not incorporated into the fitting preferences.
  8. Lines 764-766 – another explanation is that the patients population might be divided into two separate groups, one with the oxidized form and one with the native one. Have the authors detected antibodies against both forms in the same patient sera?
  9. Figure 11 – what is the y-axis?
  10. Figure 11 – it is not clear how the ELISA was performed and how the authors have interpolated the results into that graph. I suggest that they add the graphs as a supplementary data.

General comments:

  1. The authors should add to the graph legends the number of replicates used for each measurement and what are the standard error represents (SEM? SD?)
  2. The figure legends are very lengthy. In many cases they include background, discussion etc. It makes the reading very problematic and I strongly suggest shortening it to the minimum that is required to describe what is displayed in the figure.

Reviewer 2 Report

The article describes that molecular change and feature of La protein associated with autoantigen of an autoimmune disease. The results and description include many important points for understanding formation of an autoantigen.

Minor points

  1. The title is somehow unclear.
  2. line 82; Spell out ALARM NMR
  3. line 135; pro analysi?
  4. line 195~196; the meaning of the sentence is unclear.
  5. line 247; thereof?
  6. line 316; per si?
  7. line 424; Spell out DTT
  8. Is image quality throughout all the Figure OK?

Reviewer 3 Report

Excluding a very catching title and a sufficiently catching abstract, this work presents several problems:

  • the authors don't even bother to describe the Sjögren’s syndrome;
  • the legends to the figures are used as an extension to the introduction;
  • the style is not consistent in the manuscript;
  • figures are cut (see figure 5 and 9);
  • figure 10 b is missing;
  • discussion is mostly repetition of the results;

the authors also missing to say what is the relevance of their paper? 

  • methods are poorly described;
  • it was never mentioned how many times the experiments were performed.

Reviewer 4 Report

In this manuscript, Berndt and colleagues reveal that La protein is sensitive to oxidation at three different cysteine residues (one of which is a novel oxidation site). They demonstrate that this oxidation leads to conformational changes of the protein and influences the ability of the protein to form homo-dimers/oligomers. They also describe the effects this oxidation can have on antibody recognition of the protein, thus revealing a possible explanation for current controversy in the field. This is a very thorough and interesting study that could have important impacts on future studies of this protein and other proteins that may behave similarly. However, there are a few concerns.

Major concerns:

  1. The entire paper focusses on oxidation and reduction dependent changes in protein conformation and multimer formation, but the introduction does not present a background on this topic. The authors should include a paragraph discussing the importance of oxidation and reduction on protein conformational changes and complex formation.
  2. In the text discussing Figure 4 (see line 319 and others) the authors make the claim that La protein is being post translationally modified at the specified epitope. However, the authors do not demonstrate this. This might be inferred from the data, but is not proven. The authors should either prove a change in phosphorylation (i.e. using a phosphospecific antibody in their western blots with the mutant constructs) or they should soften the language in the text.
  3. The authors mention SPR data (line 387) that would provide confirmation of the western blotting data in Figure 5. This data should be included in Figure 5 and not listed as "data not shown". This is an essential confirmation of the data that is the crux of the story using a separate experimental system.
  4. In general, many of the conclusions for experiments are drawn qualitatively. The authors should include information about how many independent experiments led to the these conclusions and, where possible, should include a quantitative or semi-quantitative analysis of their data. For examples:
    1. Figure 4 should include some kind of counting (manual or computational) to quantify the % of GFP+ cells that are also detected by the antibody against LA in a larger number of cells (i.e. 50 cells per replicate, 3 replicate counts).
    2. Figure 6 should have densitometry performed and replicates averaged so that a quantitative (if not statistical) analysis can be performed between the untreated samples and the oxidized samples to confirm the authors' claim that "it becomes evident that the intensity of these higher molecular weight protein bands increases in case the samples are pretreated with CuSO4 prior to SDS-PAGE."
  5. This may just be an issue with the proofs that were generated, but I do not see a Figure 10B or any colored lines at all in Figure 10, though this is discussed in the text. Please add this figure or remove the text concerning these data.

Minor concerns:

  1. Line 262 should say Figure 4 not Figure 3.
  2. Can the authors explain or infer what the additional bands are in the reduced western blots in Figure 6C. Are these degradation products of the protein? Given the focus throughout the paper on multimer formation and deletion constructs, it would be clearer if those bands were labelled.
  3. The graphs in Figure 8 need a legend.
  4. Figure 11 needs a label for the y-axis.
  5. The wording/grammar was at times difficult to understand and could use some editing. As one example, in the legend for Figure 8 there is this sentence: "In case La protein can 599 undergo redox dependent conformational changes besides an oxidized conformation also a reduced form should exist leading to the question whether also anti-La mAbs directed to the reduced form do exist." This sentence is difficult to understand and is somewhat representative of some unclear wording throughout.

Round 2

Reviewer 3 Report

The manuscript was significantly improved, even though the legend of Figure 1 is still too long.

In general, before submitting a paper it is authors duty to check the pdf and verify that all the figures are clearly visible and all the panels are described in the legends.